# Nano3D: A Training-Free Approach for Efficient 3D Editing Without Masks

**Junliang Ye**[1*]    **Shenghao Xie**[2,1*]    **Ruowen Zhao**[1]    **Zhengyi Wang**[1]    **Hongyu Yan**[4]
**Wenqiang Zu**[5,2]    **Lei Ma**[2†]    **Jun Zhu**[1,3†]
[1]Tsinghua University  [2]Peking University  [3]ShengShu  [4]HKUST  [5]CASIA

Figure 1: **Highly-consistent 3D objects edited by Nano3D.** Our framework supports a range of training-free and part-level tasks especially on shape, including removal, addition, and replacement, while only requiring users to provide source 3D objects and instructions, without any mask.

## ABSTRACT

3D object editing is essential for interactive content creation in gaming, animation, and robotics, yet current approaches remain inefficient, inconsistent, and often fail to preserve unedited regions. Most methods rely on editing multi-view renderings followed by reconstruction, which introduces artifacts and limits practicality. To address these challenges, we propose **Nano3D**, a training-free framework for precise and coherent 3D object editing without masks. Nano3D integrates FlowEdit into TRELLIS to perform localized edits guided by front-view renderings, and further introduces region-aware merging strategies, Voxel/Slat-Merge, which adaptively preserve structural fidelity by ensuring consistency between edited and unedited areas. Experiments demonstrate that Nano3D achieves superior 3D consistency and visual quality compared with existing methods. Based on this framework, we construct the first large-scale 3D editing datasets **Nano3D-Edit-100k**, which contains over 100,000 high-quality 3D editing pairs. This work addresses long-standing challenges in both algorithm design and data availability, significantly improving the generality and reliability of 3D editing, and laying the groundwork for the development of feed-forward 3D editing models.

## 1 INTRODUCTION

Generative models for 3D asset creation have made tremendous progress Lai et al. (2025); Chen et al. (2025b; 2024e); Wang et al. (2023), leading to widespread applications across entertainment, robotics, and healthcare. In particular, recent rectified flows (reflows) Liu et al. (2022), such as TRELLIS Xiang et al. (2025), achieve high-quality 3D object generation by embedding heterogeneous representations into a unified latent space while explicitly disentangling geometry and appearance. Beyond generation, editing (*i.e.*, revising the intended region while keeping other regions unchanged) is also valuable as users usually need to refine existing assets rather than regenerate entirely new ones, which requires multiple unpredictable iterations to obtain a satisfactory result.

---

[*]Equal contribution
[†]Corresponding author.

For example, TRELLIS can generate diverse plausible appearances easily with style-modified text or image prompts, such as texture and material, but fail to reliably repeat identical geometries.

In image editing, an increasing number of powerful models have recently emerged, including GPT-4o Hurst et al. (2024), Flux.1 Kontext Labs et al. (2025), and Nano Banana Fortin et al. (2025). A closer look at the evolution of these models reveals a clear three-stage development paradigm. Stage 1 introduced training-free image editing algorithms Hertz et al. (2022), which demonstrated the feasibility of editing without model finetuning. Stage 2 focused on the automatic construction of large-scale, high-quality paired editing datasets, providing the foundation for supervised learning Brooks et al. (2023). Stage 3 leveraged these datasets to train feedforward image editing models capable of real-time inference and high fidelity generation.

In contrast, 3D object editing still remains bottlenecked in the initial stage (*i.e.*, algorithm). Specifically, existing methods, such as those based on Score Distillation Sampling (SDS) Sella et al. (2023) or the "multi-view editing then reconstruction" paradigm Qi et al. (2024), struggle to maintain consistency across views or attributes and usually demand time-consuming optimization. This leaves us wondering: *can 3D objects be edited versatilely, efficiently and consistently in a training-free manner using only pretrained generative models, as achieved in 2D images?* Resolving this problem will allow 3D object editing to enter a virtuous cycle of data expansion and training models capable of flexible asset customization, thereby accelerating the whole field toward maturity like 2D images.

We propose Nano3D, a training-free 3D editing algorithm designed for constructing paired 3D editing datasets. Drawing inspiration from the training-free 2D editing method FlowEdit Kulikov et al. (2024), Nano3D leverages the first stage of TRELLIS to generate an iterative trajectory from input to edited voxel representations, thereby enabling efficient training-free 3D editing.

To further enhance source consistency between the original and edited objects, we introduce a region-aware merging strategy, Voxel/Slat-Merge, applied after TRELLIS's two-stage geometry and appearance editing. Based on simple connectivity analysis, this strategy adaptively identifies modified voxel regions in the edited 3D object and integrates them back into the original object. This effectively merges the edited content while preserving the structure of unedited regions.

Building on the Nano3D algorithm, we design an efficient pipeline for large-scale construction of 3D editing datasets and generate a high-quality dataset of 100,000 samples——**Nano3D-Edit-100k**. Our work addresses two long-standing gaps in the 3D editing domain—the lack of training-free editing algorithms and the absence of large-scale datasets—thereby laying a solid foundation for the third stage of 3D editing: training feedforward models under 3D editing supervision.

Overall, our contributions can be summarized as follows:

- We make the first attempt to introduce FlowEdit to 3D object editing, demonstrating that the powerful priors of 3D object generative models can also support effective training-free editing (like 2D images)

- We propose Voxel/Slat-Merge, a region-aware merging strategy that automatically preserves source consistency in the non-edited regions of 3D objects.

- We develop a user-friendly 3D editing framework, **Nano3D**, which achieves state-of-the-art editing performance without requiring any manual masks.

- Building upon Nano3D, we curate the first large-scale 3D editing dataset **Nano3D-Edit-100k**, comprising over 100,000 high-quality samples to support further research and development.

## 2 RELATED WORK

### 2.1 2D IMAGE EDITING

With the advent of large-scale 2D generative models, image editing has shifted from manual pixel-level operations to controllable semantic-level manipulation. Early approaches modify noisy latents via inversion to balance new details with original structures Meng et al. (2021); Mokady et al. (2023); Abdal et al. (2019), while others finetune generative models on curated editing pairs to enable instruction following Brooks et al. (2023); Wei et al. (2024); Sheynin et al. (2024). Localized editing has also been explored through attention map manipulation Hertz et al. (2022); Tumanyan et al.

(2023); Couairon et al. (2022), and adapters have been introduced to inject additional conditions for enhanced controllability Ye et al. (2023); Ju et al. (2024); Mou et al. (2024). More recently, rectified flows (reflow) Liu et al. (2022); Esser et al. (2024) have enabled high-fidelity synthesis with few sampling steps. To support reflow-based editing, RFSolver Wang et al. (2024) approximates ODEs via higher-order Taylor expansion while preserving structures through attention replacement, whereas FlowEdit Kulikov et al. (2024) introduces an inversion-free strategy by interpolating between sampled noise and the source image.

## 2.2 3D EDITING

Compared to 2D image editing, maintaining spatial consistency is substantially more challenging in 3D. Many approaches adopt score distillation sampling (SDS) Poole et al. (2022) to optimize 3D representations using gradients from pretrained 2D diffusion models Sella et al. (2023); Li et al. (2024); Chen et al. (2024c); Palandra et al. (2024); Chen et al. (2023). Others edit multi-view images and reconstruct them with large reconstruction models (LRMs) Qi et al. (2024); Chen et al. (2024a); Barda et al. (2025); Huang et al. (2025); Erkoç et al. (2025); Bar-On et al. (2025); Zheng et al. (2025); Li et al. (2025); Gao et al. (2024), or directly manipulate triplanes as a bridge between 2D and 3D Kathare et al. (2025); Bilecen et al. (2025). Inspired by InstructPix2Pix Brooks et al. (2023), several works construct paired 3D editing datasets for supervised training Ye et al. (2025); Xu et al. (2023). To enable finer control, diverse conditions such as sketches Mikaeili et al. (2023); Liu et al. (2024); Guillard et al. (2021), part-level masks Chen et al. (2025a); Yang et al. (2025a;b); Wang et al. (2025a), and point-based dragging Chen et al. (2024b); Xie et al. (2023); Lu et al. (2025) have been explored. Beyond manipulating individual objects, extending these editing capabilities to complex 3D scenes has also garnered significant attention Chen et al. (2024d); Wang et al. (2025b). In this work, we unlock their potential for versatile and consistent 3D editing in a training-free and user-friendly manner.

## 3 PRELIMINARY

### 3.1 FLOWEDIT

FlowEdit Kulikov et al. (2024) is a text-guided image editing method tailored for text-to-image flow models. It is characterized by being inversion-free, optimization-free, and model-agnostic. Rather than relying on traditional inversion-reconstruction processes that often introduce distortion, FlowEdit constructs an ordinary differential equation (ODE) trajectory in the latent space from the source prompt to the target prompt. This trajectory enables direct evolution of image representations over the velocity field. By leveraging a weighted combination of the source and target velocity fields, FlowEdit ensures a shorter editing path and stronger structural preservation throughout the editing process. Therefore, **given a source image, along with the two conditions before and after editing (*e.g.*, the text describing the image or a single view rendered from the 3D asset), the pretrained generative models can adopt FlowEdit to output the target image**. We provide a more detailed descriptions of FlowEdit's editing process in Sec. A.13

### 3.2 TRELLIS

TRELLIS generates 3D objects through a two-stage geometry-appearance decoupling pipline. In stage 1, **it predicts a sparse structure from noise**, which represents geometry by a voxel occupancy grid $s = \{s_i\}_{i=1}^{L}$, where $s_i \in \{0, 1, \ldots, N-1\}^3$, $L$ is the grid spatial length and $N$ is the number of active voxels. In stage 2, **TRELLIS predicts a structured latent (SLat) based on** $s$, which further incorporates appearance information, represented by $z = \{(z_i, s_i)\}_{i=1}^{L}$, where $z_i \in \mathbb{R}^C$ is the aggregated multi-view DINOv2 feature for the $i$-th voxel, with $C$ as the feature dimensionality.

## 4 METHOD

### 4.1 OVERVIEW

A common approach is to edit rendered images of a 3D object and reconstruct it with a generative model, but this often breaks geometric consistency. To address this, we introduce FlowEdit into

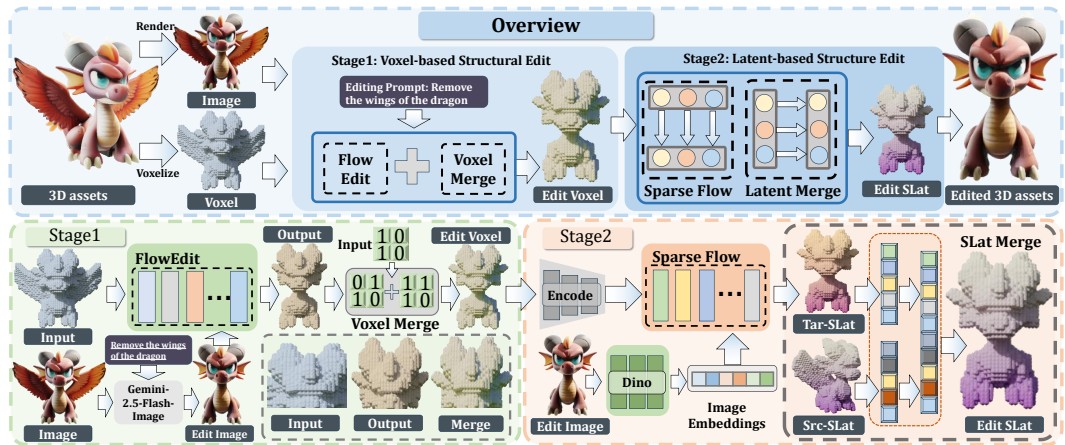

Figure 2: **The Nano3D pipeline.** The original 3D object is voxelized and encoded into sparse structure and structured latent respectively. Stage 1 modifies geometry via Flow Transformer with FlowEdit, guided by Nano Banana–edited images. Stage 2 generates structured latents with Sparse Flow Transformer, supporting TRELLIS-inherent appearance editing. Voxel/Slat-Merge further ensures consistency across both stages before decoding the final 3D object.

the first-stage generation of TRELLIS (Sec. 4.2). To further ensure geometric and appearance consistency, we propose Voxel/Slat-Merge (Sec. 4.3), which detects edited regions and integrates them with unedited ones. Finally, we present a training-free, user-friendly pipeline (Sec. 4.4) that also supports large-scale dataset construction. By combining FlowEdit with Voxel/Slat-Merge, Nano3D achieves geometrically consistent and semantically faithful 3D object editing within TRELLIS.

## 4.2 FLOWEDIT

Inspired by FlowEdit's success in 2D image editing, we extend it to 3D object editing by integrating it into TRELLIS stage 1, leveraging the pretrained generative prior to establish an editing path between source and target objects instead of starting from noise. The input is the source 3D voxel grid $s_{src}$, TRELLIS applies the FlowEdit algorithm to output the edited target voxel grid $s_{tgt}$, by treating the rendered front-view image $c_{src}$ and the modified target front-view image $c_{tgt}$ as the source control condition and the target control condition respectively. Specifically, this process is divided into the following two stages:

**Front-View Image Editing.** Given the editing instruction $txt$ and the rendered front-view image $c_{src}$ of the source 3D object, we first utilize the advanced 2D image editing model Nano Banana to edit $c_{src}$, thereby obtaining the edited front-view image $c_{tgt}$ of the target 3D object.

**Voxel Editing.** Subsequently, we consider the voxel grids $s_{src}$, $s_{tgt}$ of the source and target 3D objects, and define the noise-to-voxel generation trajectories as $p_t$, $q_t$. With $c_{src}$, $c_{tgt}$ serving as conditions, the velocity fields $v_t^\theta(p_t, c_{src})$, $v_t^\theta(q_t, c_{tgt})$ of these trajectories are predicted by the pretrained Flow Transformer from TRELLIS stage 1. Then FlowEdit is adopted to establish an editing trajectory $s_t$ with timestep $t \in [0, 1]$ by aligning $p_t$, $q_t$ to start from the same sampled noise $\epsilon \sim \mathcal{N}(0, I)$:

$$s_t = s_{\text{src}} + q_t - p_t, \tag{1}$$

$$\approx s_{\text{src}} + \left( v_t^\theta(q_t, c_{\text{tgt}}) - v_t^\theta(p_t, c_{\text{src}}) \right) dt. \tag{2}$$

$$\text{where } p_t = (1-t)s_{src} + t\,\epsilon, \quad q_t = (1-t)s_{tgt} + t\,\epsilon.$$

Such rectified flow-based trajectory gradually moves toward $s_0 = s_{tgt}$ under the semantic guidance provided by the velocity field differences, and preserves the source geometry consistency by starting from $s_1 = s_{src}$, rather than directly generates $s_{tgt}$ from a random noise.

## 4.3 Voxel/Slat-Merge

**Voxel-Merge.** We observe that the voxel edited by FlowEdit sometimes still exhibit minor geometry inconsistencies with the source 3D object, *e.g.*, when editing a dragon to remove its wings, the result may not only modify the wings but also inadvertently alter other unrelated regions. To this end, we further introduce a region-aware merging strategy, Voxel-Merge, **which takes the source 3D voxel grid $s_{src}$ and the FlowEdit-edited voxel grid $s_{fe}$ as inputs, and outputs the final target voxel grid $s_{tgt}$ that merges the desirable edited regions from $s_{fe}$ with the non-edited regions from $s_{src}$.** Specifically, it defines a difference map $g$ via an element-wise XOR operation between $s_{src}$ and $s_{fe}$:

$$g(i) = s_{src}(i) \oplus s_{fe}(i) = \begin{cases} 1, & \text{if } s_{src}(i) \neq s_{fe}(i), \\ 0, & \text{if } s_{src}(i) = s_{fe}(i). \end{cases} \quad \forall i \qquad (3)$$

where all modified elements of $s_{fe}$ are explicitly marked with 1, and connectivity analysis is employed on such elements to group them into distinct regions. Regions larger than the threshold $\tau$ are then selected, separating the desired edited regions from those irrelevant modifications. Next, a binary mask $m$ is initialized, with elements corresponding to the selected regions set to 1 and the rest to 0. Finally, another XOR operation is performed between the mask and $s_{src}$:

$$s_{src} \oplus m \rightarrow s_{tgt}. \qquad (4)$$

thereby transferring the correct edited regions onto $s_{src}$ with the non-edited regions preserved.

Now that the geometry consistency is sufficiently achieved, we proceed to feed the merged voxel grid $s_{tgt}$ together with the edited front-view image $c_{tgt}$ into TRELLIS stage 2, leveraging its pretrained Sparse Flow Transformer to output the target SLat $z_{tgt}$. To similarly ensure the generated SLat $z_{tgt}$ are consistent with $z_{src}$ encoded from the original 3D object, we also introduce SLat-Merge by reusing the mask $m$ during the Voxel-Merge stage and performing:

$$z_{src} \oplus m \rightarrow z_{tgt}^{\cdot}. \qquad (5)$$

**Therefore, SLat-Merge outputs the final merged target SLat $z_{tgt}^{\cdot}$ by combining the appearance features of both the non-edited and desirable edited regions from the input $z_{src}$ and $z_{tgt}$, preserving the appearance consistency.**

## 4.4 Nano3D

As illustrated in Fig. 2, Nano3D builds upon TRELLIS to enable decoupled geometry and appearance editing of 3D objects. The input object is voxelized and, along with DINOv2 Oquab et al. (2023) features, encoded into a structured latent representation via a VAE Kingma & Welling (2013). Meanwhile, we use Nano Banana with the front view of a 3D asset and editing instructions as input to generate the edited front view. In TRELLIS-Stage 1, we replace the standard flow iteration with FlowEdit, which takes the source object's voxel and the before/after front views as input, and outputs the edited voxel. We then apply Voxel-Merge to ensure geometric consistency. In TRELLIS-Stage 2, the edited voxel and edited front view jointly guide TRELLIS to generate the final SLat. At this stage, we further adopt Slat-Merge to guarantee both geometric and texture consistency. Finally, the edited SLat is decoded by the VAE to reconstruct the target 3D object.

**Data Construction Pipeline.** As illustrated in Fig. 3, we extend **Nano3D** by constructing a complete and streamlined **3D editing data generation pipeline**. The process consists of the following stages:

1. **Image Sampling from Existing Datasets**: We sample views from publicly available 3D asset datasets Xiang et al. (2025); Deitke et al. (2022). For each asset, the frontal view is selected as the editing target.

2. **Instruction Generation via VLM**: An editing instruction is automatically generated using the vision-language model **Qwen-VL-2.5** Bai et al. (2025), based on three predefined prompt templates:

   - **Add**: `add <something> to <somewhere>`

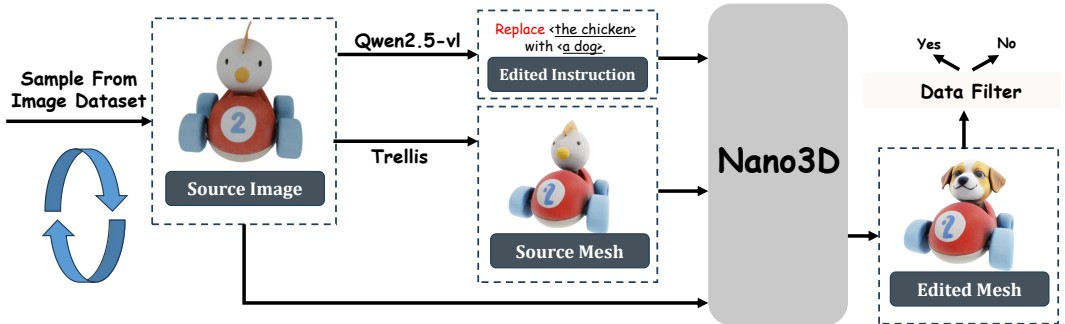

Figure 3: **Data Construction Pipeline.** The figure shows our pipeline. We first sample images from the dataset and prompt Qwen2.5-VL to generate editing instructions by completing templates. Trellis then generates 3D meshes from the images. Finally, the image, instruction, and mesh are fed into Nano3D, and the resulting 3D assets are filtered for quality.

- **Remove**: `remove <something> in <somewhere>`
- **Replace**: `replace <something> with <something>`

The model fills in these templates with visual context from the image to produce diverse and semantically grounded instructions.

3. **3D Asset Generation via TRELLIS**: Given the selected image, we use **TRELLIS** to reconstruct the corresponding 3D asset. Instead of using the original mesh, we choose to regenerate the source mesh via TRELLIS for two reasons: (1) obtaining the structured latent (sLat) from the original mesh requires rendering ~150 views, which is inefficient; (2) the reconstructed sLat still diverges from the original mesh due to the inherent loss in TRELLIS's VAE encoding. Using the TRELLIS-reconstructed mesh ensures consistency and reduces computational overhead.

4. **Image Editing via Nano-Banana or Flux-Kontext**: The generated instruction is input into **Nano-Banana** or **Flux-Kontext** to synthesize the edited target image.

5. **3D Editing via Nano3D**: The original 3D asset, the source image, and the edited image are fed into **Nano3D**, which outputs an edited 3D asset.

## 5 EVALUATION

### 5.1 SETUP

**Implementation Detail.** Our method is implemented on TRELLIS. The sampling step is fixed at 25, and FlowEdit is configured with $n_{max} = 15$, $n_{min} = 0$, and $n_{avg} = 5$. The CFG guidance scales for $v_t^\theta(p_t)$ and $v_t^\theta(q_t)$ are set to 1.5 and 5.5, respectively, with $\lambda$ set to 0.5. For both Voxel-Merge and Slat-Merge, $\tau$ is set to 100. For the construction of Nano3D-Edit-100k, we employ 32 A800 GPUs for inference, utilizing the Qwen2.5-vl-72B API to generate editing instructions and Flux-Kontext to perform image editing operations. The creation of each editing pair required approximately five minutes, and empirical observations revealed two key findings: first, the vast majority of failed cases originated from errors in the image editing stage, whereas successful adherence to instructions at this stage led to a very high success rate in the subsequent Nano3D editing process; second, the predominant computational cost arose from the Flexicube module, which consumed nearly 4.5 minutes per pair, while the preceding steps required only about 30 seconds. Based on these observations and in order to further reduce computational overhead, we adopted a storage strategy in which only the SLAT (Structured Latent) representation and the voxel sum of each asset are preserved, thereby allowing users to flexibly decide whether to directly train on SLAT or to employ Flexicube to convert SLAT into explicit GLB meshes for downstream applications. To improve dataset quality, we use Qwen2.5-VL-7B to automatically filter edited images based on instruction compliance. Non-compliant samples are returned to the pool for re-sampling.

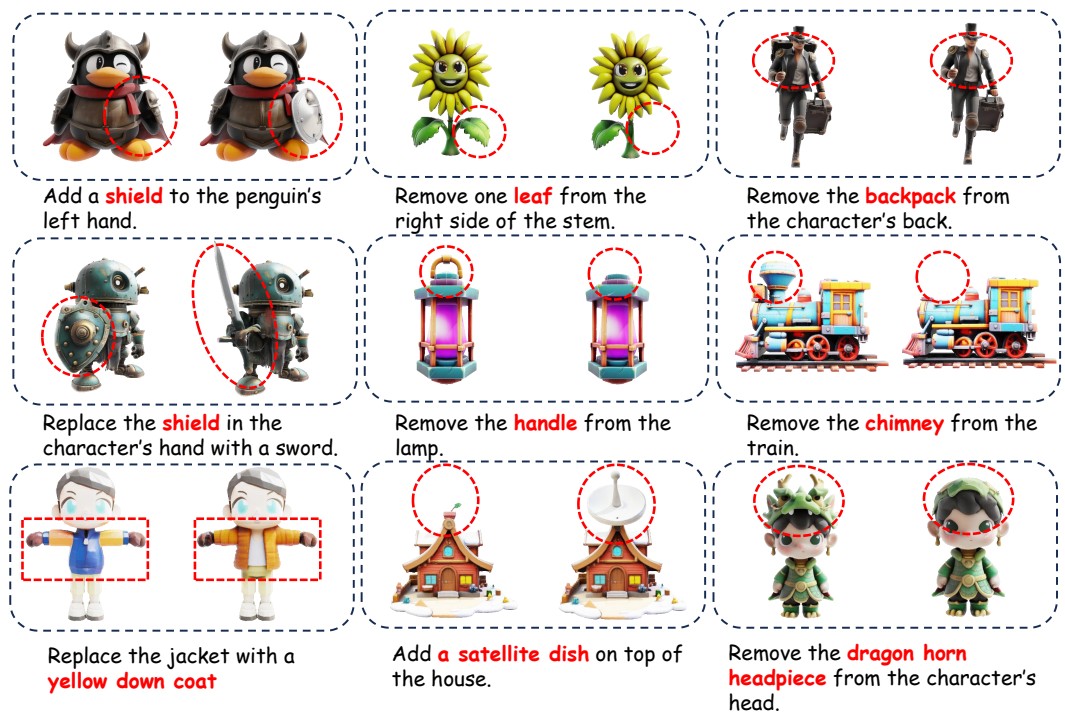

Figure 4: **Qualitative results.** We present three edit types—object removal, addition, and replacement. In each case, Nano3D confines changes to the target region (red dashed circles) and produces view-consistent edits, while leaving the rest of the scene unchanged. Geometry stays sharp and textures remain faithful in unedited areas, with no noticeable artifacts.

**Baseline.** We select three representative state-of-the-art methods as baselines: Vox-E based on SDS, Tailor3D based on "multi-view editing then reconstruction", and TRELLIS, which leverages a RePaint-based method. For all baselines, we strictly follow their original implementations and use the official codebases to obtain the results reported in this paper.

**Dataset.** Our **Nano3D-Edit-100k** dataset comprises two sources of image data: images collected from the internet and rendered views from the Trellis-500K dataset. During dataset construction, we follow the methodology of 3D-Alpaca Ye et al. (2025), employing Qwen2.5-VL to automatically annotate 3D assets and classify them accordingly. We then perform class balancing across ten distinct categories, ultimately selecting 100K image samples. We select 100 representative cases from the Nano3D-Edit-100k dataset for the experiments and demonstrations in this section.

**Metric.** We systematically evaluate the edited 3D objects from three perspectives: source structure preservation, target semantic alignment, and generation quality. For source structure preservation, we assess non-edited regions against the original 3D object using Chamfer Distance (CD) Fan et al. (2017). For target semantic alignment, we employ the DINO-I Caron et al. (2021) metric to quantify adherence to the target edited image. For generation quality, we use FID Heusel et al. (2017) on rendered multi-view images to measure fidelity and diversity.

## 5.2 MAIN RESULT

**Qualitative Comparison.** As shown in Fig. 5, Nano3D not only strictly follows editing instructions but also maintains perfect structure consistency with the source 3D object across multi-view images. In contrast, Tailor3D introduces noticeable geometry distortions and appearance artifacts. Vox-E produces results that are overly blurry, smoothed, and misaligned with the target semantic. TRELLIS, though showing relative improvements, still suffers from several issues, such as local detail corruption, shape enlargement, and incorrect orientation. These findings demonstrate that our method delivers impressive and steady visual effects beyond the reach of existing methods.

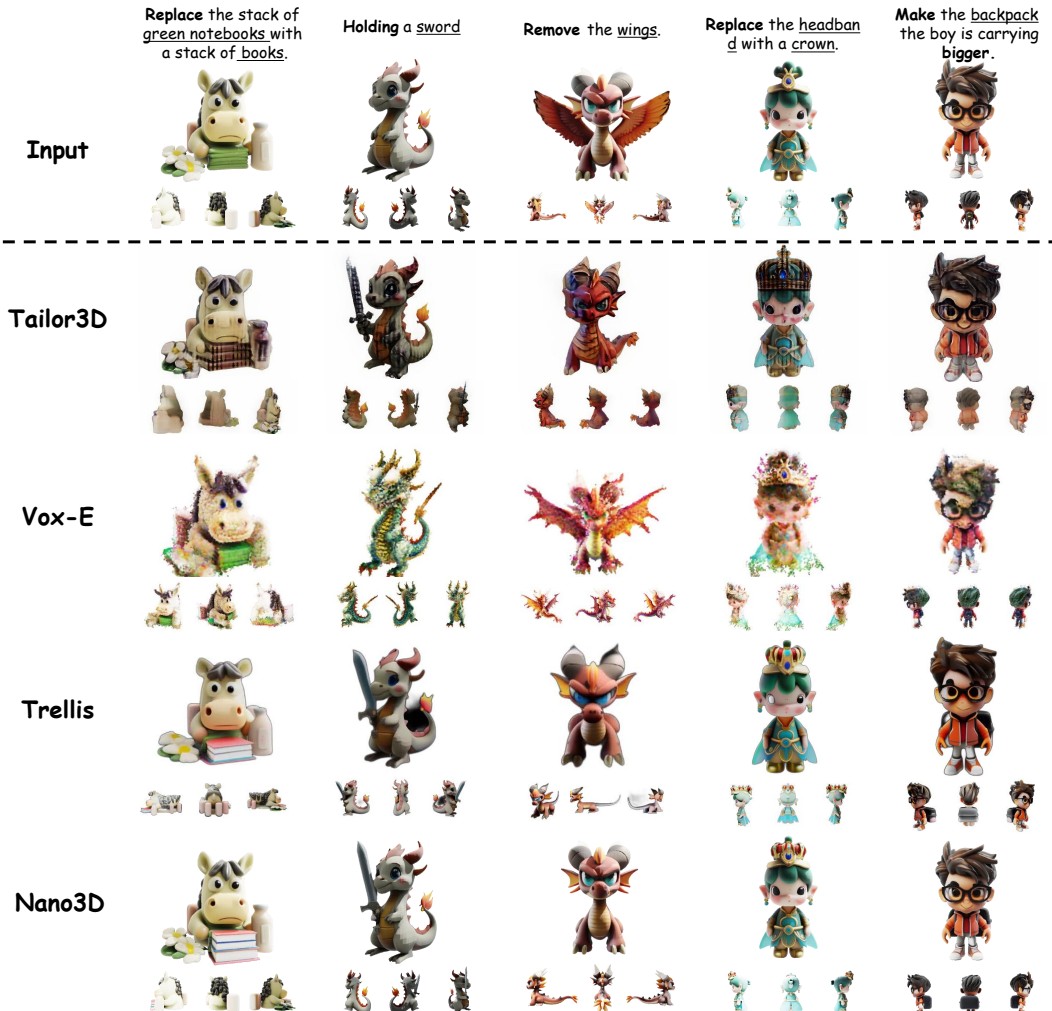

Figure 5: **Qualitative comparison.** Our method achieves the best editing performance with faithful instruction semantic alignment and perfect original structure consistency across multi-view images.

Table 1: **Quantitative comparison.** Our method achieves the best structure consistency, semantic alignment with the target edited image, and generation fidelity.

| Method | CD↓ | DINO-I↑ | FID↓ |
|---|---|---|---|
| Tailor3D | 0.037 | 0.759 | 140.93 |
| Vox-E | / | 0.782 | 117.12 |
| TRELLIS | 0.019 | 0.901 | 49.57 |
| Instant3DiT | 0.014 | 0.879 | 56.73 |
| Nano3D | **0.013** | **0.950** | **27.85** |

Table 2: **User study.** Given that most users favored TRELLIS and Nano3D, the results for Tailor3D and Vox-E are omitted from the table for clarity. As shown in the table, our method is strongly preferred by participants, significantly outperforming TRELLIS.

| Method | Prompt Algn. | Visual Quality | Shape Preserv. |
|---|---|---|---|
| TRELLIS | 32% | 21% | 5% |
| Nano3D | 68% | 79% | 95% |

**Quantitative Comparison.** As shown in Tab. 1, Nano3D outperforms all baselines, achieving the lowest CD and FID and the highest DINO-I score, indicating superior structural consistency, perceptual quality, and semantic alignment, as seen in Fig. 5.

**User Study.** To assess editing quality and usability, we conducted a user study with 50 participants. Each round presented the original 3D object, task instructions, and results from Tailor3D, Vox-E, TRELLIS, and Nano3D. Participants selected the best method based on Prompt Alignment, Visual

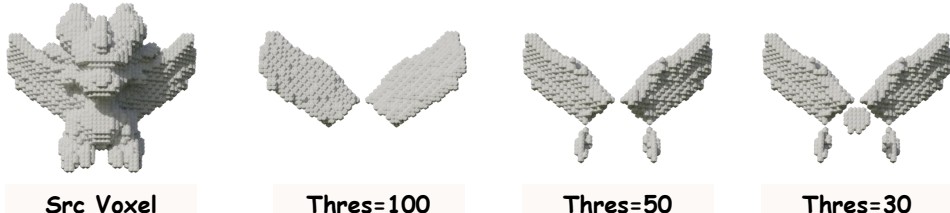

Figure 6: **Ablation study on** $\tau$**.** The leftmost voxel represents the pre-editing state, with the editing instruction being to remove the wings. The three voxels on the right correspond to the masks generated during the voxel-merge stage for $\tau = 100$, $\tau = 50$, and $\tau = 30$ (from left to right). As observed, when $\tau = 100$, the detected mask most accurately aligns with the editing regions, while lower values include irrelevant non-editing areas.

Quality, and Shape Preservation. As shown in Tab. 2, Nano3D received the highest preference across all criteria, demonstrating superior semantic alignment, visual quality, and shape fidelity. For clarity, Tailor3D and Vox-E results are omitted, as user choices mainly favored TRELLIS and Nano3D.

**Nano3D-Edit-100k v.s. 3D-Alpaca.** High-quality 3D editing requires consistency in both 2D image appearance and 3D structure—that is, the rendered images before and after editing should remain coherent, and the 3D assets themselves should preserve structural integrity throughout the editing process. The 3D-Alpaca dataset lacks both aspects, leading to significantly lower data quality compared to ours. To quantify this, we randomly sample 500 edited pairs from each dataset and evaluate text–image alignment using CLIPScore Hessel et al. (2021) and ViLT R-Precision Kim et al. (2021). Specifically, we use a VLM to infer the caption of the edited asset based on the original asset's caption and the editing instruction. As shown in Table 3, our **Nano3D-Edit-100k** consistently outperforms 3D-Alpaca across all metrics.

Table 3: Semantic alignment comparison between NANO3D-EDIT-100K and 3D-Alpaca.

|  | CLIPScore | ViLT R-Precision R@5 | ViLT R-Precision R@10 |
|---|---|---|---|
| **3D-Alpaca** | 28.42 | 33.6 | 40.2 |
| **Nano3D-Edit-100k** | **39.71** | **45.3** | **52.4** |

### 5.3 ABLATION STUDY

**Voxel/Slat-Merge.** We sequentially validate the effectiveness of Voxel-Merge and Slat-Merge strategies. As shown in Fig. 7, relying solely on FlowEdit leads to geometry misalignments and deformations, accompanied by missing, blurred, and distorted appearances, resulting in obvious inconsistencies with the original 3D object. Incorporating Voxel-Merge substantially improves the overall performance, restoring geometry and enhancing cross-view global consistency, but leaving appearance issues unresolved. With the additional incorporation of Slat-Merge, local visual quality is further enhanced, and appearances exhibit greater consistency before and after editing. These results indicate that our methods effectively exploit the advantage of geometry-appearance decoupling in 3D objects, ensuring more reliable consistency.

**Ablation on** $\tau$**.** We further compare different values of $\tau$, as shown in Fig. 6. The leftmost voxel represents the pre-editing state, with the editing instruction being to remove the wings. The three voxels on the right correspond to the masks generated during the voxel-merge stage for $\tau = 100$, $\tau = 50$, and $\tau = 30$ (from left to right). As observed, when $\tau = 100$, the detected mask most accurately aligns with the editing regions, while lower values include irrelevant non-editing areas.

## 6 CONCLUSION

In this work, we present **Nano3D**, a training-free and user-friendly framework for localized 3D object editing, supporting operations such as object removal, addition, and replacement. By integrating FlowEdit into the TRELLIS pipeline and introducing region-aware merging strategies (Voxel/Slat-Merge), Nano3D achieves geometrically consistent and semantically faithful edits. Extensive ex-

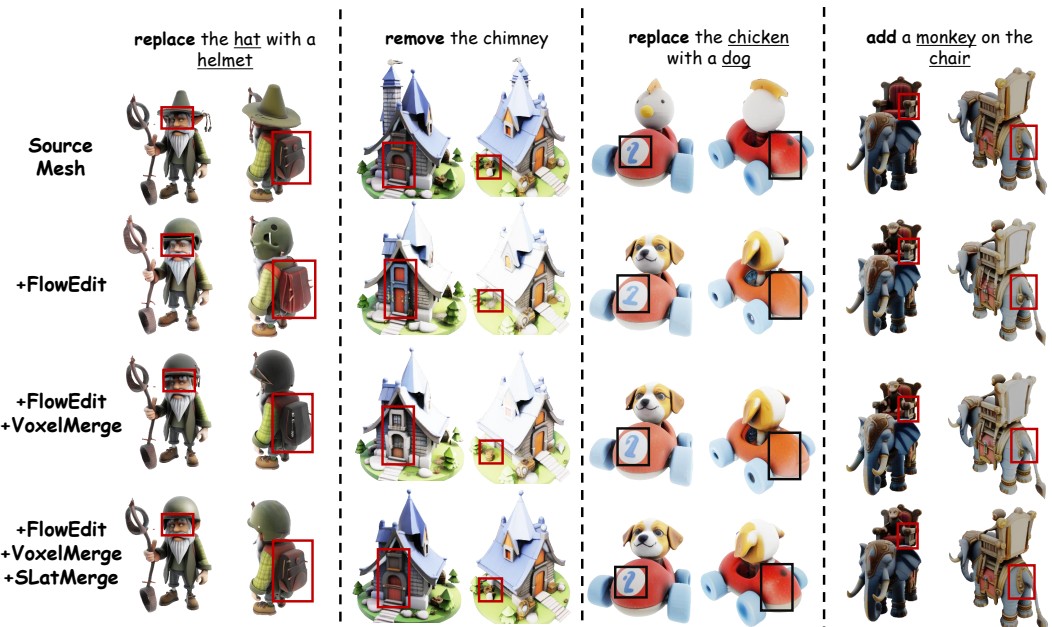

Figure 7: **Ablation study on Voxel/SLat-Merge.** Our methods sequentially ensure geometry and appearance consistency, demonstrating their complementary roles.

periments demonstrate its state-of-the-art performance across diverse editing tasks. Furthermore, we construct Nano3D-Edit-100k, the first large-scale dataset tailored for 3D editing, enabling future research on feedforward DiT-based editing models.

**Limitation.** Nano3D demonstrates strong performance in 3D editing tasks, but has the following limitations: it supports only localized edits; the VAE in TRELLIS introduces reconstruction loss; and the overall performance is constrained by TRELLIS's generative capacity. We view these limitations as important directions for future research.

## 7 ETHICS STATEMENT

We propose Nano3D, a training-free framework for precise and coherent 3D object editing without masks. Building on this framework, we construct the first large-scale 3D editing dataset, Nano3D-Edit-100k, which contains over 100,000 high-quality 3D editing pairs. This dataset is built upon publicly available data and will be released as an open resource for the research community. We declare no competing interests.

## 8 REPRODUCIBILITY STATEMENT

This paper presents a training-free 3D editing pipeline and, based on it, constructs a 100K-scale 3D editing dataset. We will release all code, annotation scripts, and the dataset as open-source resources.

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

## A APPENDIX

### A.1 LLM USAGE STATEMENT

The authors used ChatGPT-5 exclusively for grammar checking and language polishing of the manuscript text. All technical content, experimental design, data analysis, and scientific conclusions are the original work of the authors. The LLM was not involved in generating scientific ideas, conducting experiments, or interpreting results.

### A.2 TEXTURE EDITING

As shown in the Figure 8, we include an experiment on texture editing. Specifically, we render a front-view image of the source mesh and perform texture edits on the rendered image using an image editing model Labs et al. (2025). We then feed the edited image together with the original mesh into Trellis to produce textured Gaussians. Finally, we bake the resulting texture onto the input mesh. This procedure enables texture editing while preserving geometric consistency.

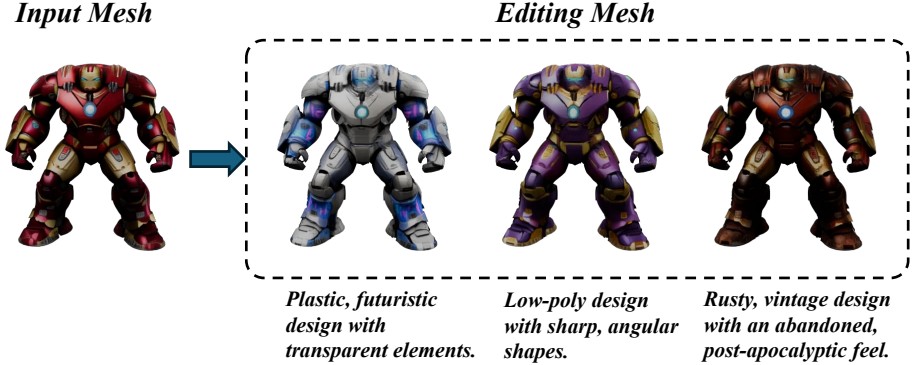

Figure 8: Nano3D can perform texture editing without altering the geometry.

### A.3 DEFORMATION EDITING

As shown in the Figure 9, Nano3D supports deformation-based editing while preserving overall consistency.

### A.4 SCENE EDITING

As shown in Figure 10, Nano3D successfully handles simple scene edits, with indoor examples on the left and outdoor examples on the right. For complex scenes, however, Nano3D alone is insufficient because the SLAT representation in TRELLIS cannot encode high-complexity inputs. To address this limitation, we adopt a block-wise editing strategy. We partition a complex scene into uniform blocks, each constrained to a level of complexity that TRELLIS can encode. We then identify the block containing the target region and apply Nano3D only to that portion. Using this strategy, we are able to edit an entire city-scale scene. As shown in Figure 11, we replace a building with a garden and subsequently replace the fountain within the garden with a sculpture. These results demonstrate that our Nano3D supports coherent and effective scene-level editing.

### A.5 ABLATION STUDY ON FRONT-VIEW SELECTION IN NANO3D

To investigate how the choice of front-view rendering influences Nano3D's editing performance, we conduct the ablation study illustrated in the Figure 13. Starting from a mesh with a fixed pose, we render four distinct viewpoints and apply Nano3D editing to each view independently. The results demonstrate that Nano3D produces plausible edits even when the input is a fully rear-facing view

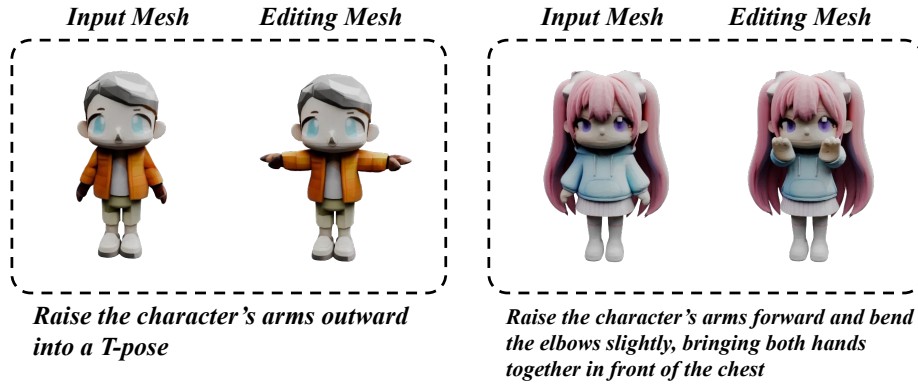

**Figure 9:** Nano3D is capable of deformation; it can modify a character's pose as required while maintaining consistency.

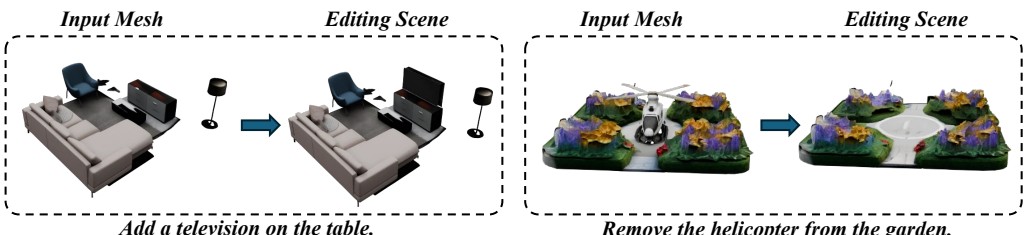

**Figure 10:** Two examples of scene editing using Nano3D are shown: the left illustrates indoor scene editing, while the right illustrates outdoor scene editing.

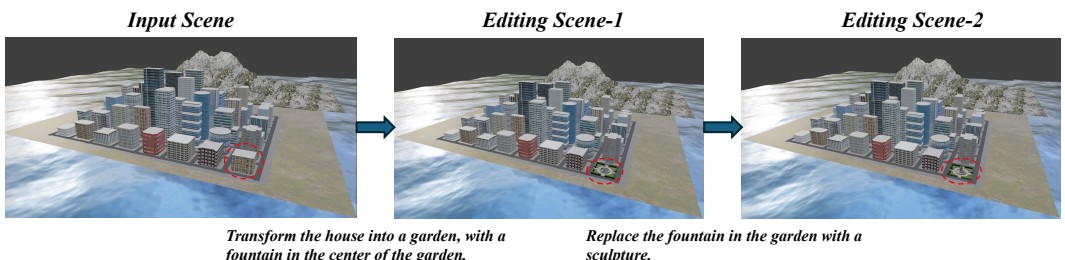

**Figure 11:** Two examples of block-wise complex scene editing using Nano3D are presented. In the left example, a building within the city is replaced with a garden featuring a central fountain. In the right example, the fountain in the garden is replaced with a sculpture.

(second to last row), despite a minor reduction in consistency. For the remaining three side-oriented views, the method yields highly coherent and consistent outcomes. These observations indicate that our method is robust to variations in the input viewpoint.

## A.6 MULTIVIEW 3D-FLOWEDIT

To address the challenges posed by complex occlusions and asymmetric objects, we incorporate a multi-view guidance mechanism into the FlowEdit pipeline. Specifically, we independently perform image editing on both the frontal and rear views of the input 3D assets. These edited images are then jointly injected into FlowEdit as guidance signals. As shown in Figure 12, this strategy significantly enhances the robustness of Nano3D. For instance, the system successfully achieves simultaneous editing of spatially distinct features—such as a fist on the front and a backpack on the back—thereby condensing a conventional two-stage workflow into a single-pass process. Consequently, this approach substantially improves the overall robustness of Nano3D.

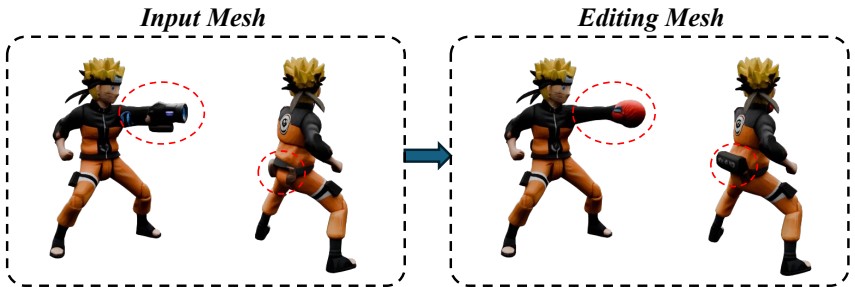

*Input Mesh*        *Editing Mesh*

*Replace the weapon in the character's right hand with a red
boxing glove, and add a waist bag on the character's back.*

Figure 12: Illustration of the multi-view guidance mechanism in Nano3D. To handle complex occlusions and asymmetric objects, we independently edit the frontal and rear views of the input 3D asset and inject them jointly into the FlowEdit pipeline. This strategy enables simultaneous editing of spatially distinct features—such as adding a fist on the front and a backpack on the back—in a single-pass process, significantly enhancing the system's robustness.

## A.7 ADAPTIVE THRESHOLD ESTIMATION

To address the limitation where a fixed threshold $\tau = 100$ fails to capture regions for fine-grained edits—such as the subtle transformation of human ears into elf ears—we propose an **Adaptive Threshold Estimation** algorithm. This method automatically adjusts $\tau$ according to the granularity of the specific editing task.

Specifically, our algorithm dynamically determines the optimal threshold during mask generation. We analyze the connected components of the potential edit regions; if the areas of all connected components are smaller than the current threshold, we iteratively halve the threshold value ($\tau \leftarrow \lfloor \tau/2 \rfloor$). This process repeats until at least one connected component exhibits an area greater than or equal to $\tau$, ensuring that relevant editing regions are captured.

As illustrated in Figure 14, we demonstrate this using Nano3D to replace human ears with elf ears. With a fixed threshold of $\tau = 100$, the algorithm fails to generate a mask because the areas of the largest connected components are $[33, 30, 13, 6, 6, 6]$, all of which fall below 100. This results in an unchanged output. In contrast, our adaptive approach detects this discrepancy and iteratively reduces $\tau$ from 100 to 50, and subsequently to 25. At $\tau = 25$, the regions corresponding to the ears (areas 33 and 30) are successfully identified, generating an accurate edit mask.

In summary, this adaptive strategy effectively extends Nano3D's capability to handle fine-grained structural changes. By eliminating the need for manual hyperparameter tuning, it ensures robust performance across editing tasks of vastly different scales.

## A.8 ADDITIONAL EVALUATION RESULTS FOR THE EDITING DATASET

In this section, we provide a comprehensive analysis of the Nano3D-Edit-100k dataset statistics and a fine-grained performance evaluation. This analysis covers the distribution of editing types, the complexity of editing instructions, and per-category semantic alignment metrics.

**Distribution of Edit Types.** To ensure the model's robustness across diverse editing scenarios, we curated the dataset to maintain a balanced composition of editing operations. As shown in Table 4, the dataset comprises approximately 40% addition instructions, with removal and replacement operations each accounting for 30%. This balanced distribution prevents the model from overfitting to specific editing patterns.

**Instruction Complexity Analysis.** We assess the difficulty of the natural language instructions to understand the dataset's complexity profile. To ensure an objective and scalable assessment, we employed a Vision-Language Model (VLM) as an evaluator. The VLM was prompted to score the

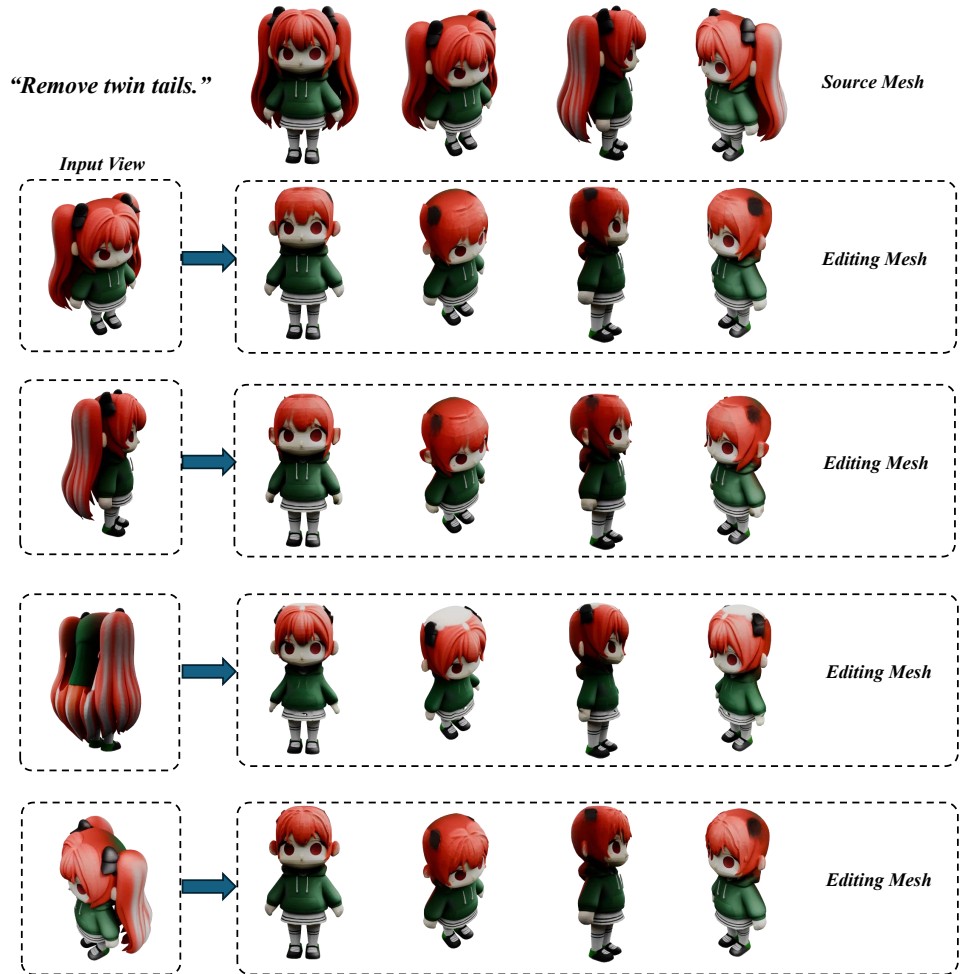

Figure 13: **Ablation study on the impact of front-view selection in Nano3D.** The results demonstrate that Nano3D produces plausible edits even when the input is a fully rear-facing view (second to last row), despite a minor reduction in consistency. For the remaining three side-oriented views, the method yields highly coherent and consistent outcomes. These observations indicate that our method is robust to variations in the input viewpoint.

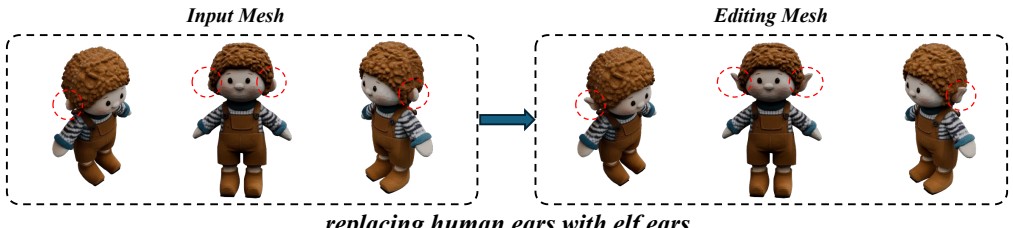

Figure 14: An example of our adaptive threshold estimation algorithm is shown in the figure, where the task is to replace human ears with elf ears. The results indicate that, with the incorporation of this algorithm, Nano3D can adaptively handle editing tasks exhibiting large variations in spatial scale, without requiring any manual hyperparameter tuning.

difficulty of each instruction on a scale from 0 (simplest) to 5 (most complex) using the following prompt:

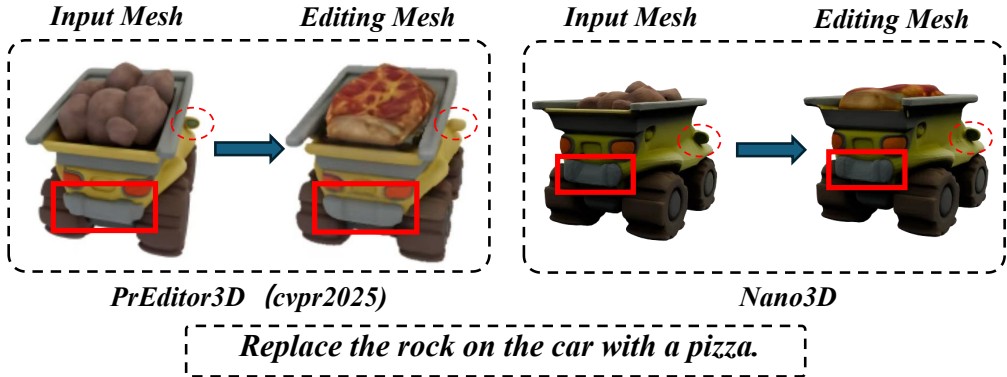

Figure 15: We conduct a visual comparison between Nano3D and PrEditor Erkoç et al. (2025). Given that PrEditor is not open-source, we utilize the results displayed on its official project page for this comparison. As illustrated in the figure, PrEditor exhibits a noticeable lack of multi-view consistency (highlighted by red boxes), which is particularly evident in the rear of the car and the side mirrors. In contrast, our method achieves highly consistent editing results. These observations demonstrate that Nano3D significantly outperforms the baseline in preserving global coherence.

Table 4: Distribution of editing types in the Nano3D-Edit-100k dataset. The dataset maintains a balanced distribution across three primary editing operations: adding, removing, and replacing objects.

| Edit Type | Count | Share (%) |
|---|---|---|
| Add | 40,000 | 40.0 |
| Remove | 30,000 | 30.0 |
| Replace | 30,000 | 30.0 |
| **Total** | **100,000** | **100.0** |

*"You are an evaluator for 3D object editing tasks. Assess the difficulty of the following instruction on a scale from 0 to 5: Instruction: {instruction}. Output strictly only the single digit."*

The resulting distribution is presented in Table 5. The statistics indicate that the majority of instructions (53.87%) fall within the moderate difficulty range (Score 3). The distribution also retains a significant portion of simpler instructions (Scores 0-2) and complex instructions (Score 4) to simulate real-world usage scenarios, while extreme cases (Score 1 and 5) remain rare.

**Per-Category Semantic Alignment.** Global metrics often mask category-specific performance nuances. Therefore, we report the fine-grained semantic alignment results in Table 6.

**Metric Note:** It is important to emphasize that the **ViLT R@5** and **R@10** metrics in this table are calculated via *intra-class retrieval* (retrieving the target from candidates within the same category). This presents a significantly more challenging fine-grained discrimination task compared to the global evaluation metrics reported in the main text (Global R@5: 45.3), where candidates are drawn from the entire dataset.

**Analysis:** The results reveal that our method achieves highly stable and robust performance on object-centric categories, such as *Food* and *Personal Item*. Conversely, performance is relatively lower on categories characterized by high visual ambiguity or structural complexity, such as *Weapon*, *Building*, and *Plant*. These findings highlight the challenge of fine-grained semantic alignment in complex structural domains and suggest meaningful directions for future optimization.

Table 5: Distribution of scores in the dataset (Total $N = 100,000$). The majority of samples fall into Score 3, while extreme scores (1 and 5) are rare.

| Score | Count | Percentage (%) |
|-------|-------|----------------|
| 0 | 12,982 | 12.98 |
| 1 | 56 | 0.06 |
| 2 | 14,674 | 14.67 |
| 3 | 53,870 | 53.87 |
| 4 | 18,390 | 18.39 |
| 5 | 28 | 0.03 |
| **Total** | **100,000** | **100.00** |

Table 6: Reports the per-category semantic alignment evaluation. It is worth noting that the R@5 and R@10 metrics in this table are calculated within each category (intra-class retrieval), which presents a more challenging fine-grained discrimination task compared to the global evaluation metrics (R@5: 45.3) reported earlier.

| | | Semantic Alignment Metrics | | |
|---|---|---|---|---|
| **Category** | **Count** | **CLIPScore** | **ViLT R@5** | **ViLT R@10** |
| Human | 20,755 | 40.17 | 34.2 | 44.4 |
| Weapon | 11,021 | 35.05 | 14.8 | 20.2 |
| Furniture | 10,442 | 39.12 | 33.6 | 45.4 |
| Personal Item | 10,277 | 38.70 | 39.0 | 52.2 |
| Animal | 10,186 | 39.59 | 32.4 | 42.6 |
| Vehicle | 9,376 | 38.47 | 28.4 | 36.6 |
| Building | 9,005 | 37.56 | 26.6 | 35.6 |
| Electronic Device | 5,283 | 38.52 | 29.6 | 39.2 |
| Plant | 4,441 | 38.67 | 29.0 | 37.8 |
| Food | 3,622 | 39.61 | 45.6 | 57.0 |

## A.9 MORE VISUALIZATION RESULTS

In Fig. 16, we showcase additional editing results covering three types of operations: addition, removal, and replacement. As illustrated in the figure, our method, Nano3D, effectively preserves both geometric and textural consistency of the 3D assets before and after editing.

## A.10 CHOICE OF 3D REPRESENTATION: VOXEL VS. VECSET

To investigate the compatibility of FlowEdit with different 3D representations, we conducted comparative experiments using Hunyuan2.1 Hunyuan3D et al. (2025), which utilizes a VecSet-based representation Zhang et al. (2023). Our analysis reveals a fundamental incompatibility between the global nature of VecSet and the localized editing requirements of FlowEdit.

Specifically, we observed that VecSet representations tend to entangle geometry globally; modifying a local region via flow matching often disrupts the overall structural integrity, leading to fragmented or collapsed geometries. In contrast, the voxel-based representation employed in TRELLIS naturally supports spatially localized modifications. The voxel grid allows FlowEdit to manipulate specific regions without propagating gradients to unrelated areas, thereby preserving the structural fidelity of the unedited parts. This structural decoupling makes voxel representations significantly more suitable for mask-free localized editing tasks.

## A.11 IMPACT OF GUIDANCE CONSISTENCY ON 3D EDITING

The quality of training-free 3D editing is heavily dependent on the multi-view consistency of the 2D guidance images. We analyze this dependency by evaluating FlowEdit on datasets with varying levels of 2D consistency, such as 3D-Alpaca Ye et al. (2025).

As illustrated in Figure 17, inconsistent 2D renderings (e.g., discrepancies in object scale or position between pre- and post-edit views) pose significant challenges. When the guidance images lack spatial alignment, the flow trajectories ($p_t$ and $q_t$) diverge, causing the editing algorithm to fail in establishing a coherent velocity field. Under such conditions, the model either ignores the editing instruction (dominance of the source structure) or produces geometric artifacts (complete loss of structure), depending on the magnitude of the noise injection ($n_{max}$). These findings underscore the necessity of our Nano3D pipeline, which ensures rigorous 2D-3D alignment prior to the editing stage.

### A.12 THE PROMPT USED TO GENERATE EDITING INSTRUCTION FROM THE RENDERING

As shown in the Table. 7, we present an example of constructing editing instructions with a VLM. A strict template is used to constrain the VLM and prevent it from generating instructions beyond Nano3D's capabilities.

### A.13 FLOWEDIT

Given a source image $x_{src} \sim X_{src}$ and target image $x_{tgt} \sim X_{tgt}$ with corresponding conditions $c_{src}, c_{tgt}$, their flow matching-based Lipman et al. (2023) generative trajectories $p_t$, $q_t$ are respectively defined as

$$p_t = (1-t)x_{src} + t\epsilon_{src}, \tag{6}$$
$$q_t = (1-t)x_{tgt} + t\epsilon_{tgt}. \tag{7}$$

where $t \in [0,1]$ is the timesteps, $\epsilon_{src}, \epsilon_{tgt} \sim \mathcal{N}(0, I)$ are the randomly sampled noise, and these trajectories are differentiated with $t$ to obtain the velocity fields $v_t(p_t, c_{src})$, $v_t(q_t, c_{tgt})$:

$$v_t(p_t, c_{src}) = \frac{dp_t}{dt} = \epsilon_{src} - x_{src}, \tag{8}$$

$$v_t(q_t, c_{tgt}) = \frac{dq_t}{dt} = \epsilon_{tgt} - x_{tgt}. \tag{9}$$

where the trajectories infer the images reversely and gradually by integrating the velocity fields from noise. In practice, the real velocity fields $v_t(p_t, c_{src})$, $v_t(q_t, c_{tgt})$ cannot be computed directly. Existing approaches convert them to conditional velocity fields Lipman et al. (2023) and train a model $\theta$ to predict $v_t^\theta(p_t, c_{src})$, $v_t^\theta(q_t, c_{tgt})$.

Unlike the generation from noise to image, FlowEdit directly defines an editing trajectory $x_t$ from the source image to target image by first aligning the starting noise of their generative trajectories (*i.e.*, $\epsilon = \epsilon_{src} = \epsilon_{tgt}$), which is based on the assumption that most regions of the source and target images are same, except for the edited regions. Then, we reformulate by combining Eq. 6 and Eq. 7:

$$x_t = x_{src} + q_t - p_t. \tag{10}$$

where $x_1 = x_{src}$ and $x_0 = x_{tgt}$, and its velocity field is

$$v_t = \frac{dx_t}{dt} \approx v_t^\theta(q_t, c_{tgt}) - v_t^\theta(p_t, c_{src}). \tag{11}$$

Therefore, this editing trajectory starts from the source image at $t = 1$ and gradually moves toward the target image at $t = 0$, guided by the semantic provided the velocity field differences in Eq. 11. At each $t$, the computation of $v_t$ requires $v_t^\theta(p_t, c_{src})$, $v_t^\theta(q_t, c_{tgt})$, and predicting them further depends on $p_t$, $q_t$, which is usually obtained by inverting the source image to the $t$-th timestep Wang et al. (2024). However, the additional inversion steps incur significant time cost. In contrast, FlowEdit shifts to constructing $p_t$ using the forward process in Eq. 6, and computes $q_t$ using Eq. 10:

$$p_t = (1-t)x_{src} + t\epsilon_t, \tag{12}$$
$$q_t = x_t + p_t - x_{src}. \tag{13}$$

where $\epsilon_t \sim \mathcal{N}(0, I)$ is the sampled noise at the $t$-th timestep. This substantially improves the efficiency of the editing process, and since both the computation of $q_t$, $v_t^\theta(q_t, c_{tgt})$ start from the intermediate latent of source image rather than the original noisy latent used in generation, the edited target image retains the structure consistency with source image.

The complete editing process is assembled in Alg. 1. In this paper, we leverage FlowEdit for 3D object editing, transforming the edited entity from the image $x$ to voxel $s$.

**Algorithm 1** Sampling mode of FlowEdit

> **Input:** $x_{\text{src}}, c_{\text{src}}, c_{\text{tgt}}$
> **Output:** $x_{\text{tgt}}$
> **Init:** $x_1 \leftarrow x_{\text{src}}$
> **for** $t \leftarrow 1$ to $0$ **do**
>     $\epsilon_t \sim \mathcal{N}(0, I)$
>     $p_t \leftarrow (1 - t)x_{\text{src}} + t\epsilon_t$
>     $q_t \leftarrow x_t - x_{\text{src}} + p_t$
>     $v_t^\theta(p_t, c_{src}) \leftarrow Model_\theta(p_t, c_{src}, t)$
>     $v_t^\theta(q_t, c_{tgt}) \leftarrow Model_\theta(q_t, c_{tgt}, t)$
>     $v_t \leftarrow v_t^\theta(q_t, c_{tgt}) - v_t^\theta(p_t, c_{src})$
>     $x_{t-1} \leftarrow x_t + v_t dt$
> **Return:** $x_{\text{tgt}} \leftarrow x_0$

Table 7: The prompt used to generate editing instruction from the rendering

| Editing Action | Prompt |
|---|---|
| Replace | Given an image, generate a short "replace" type editing instruction in the format: Replace [original object/part/pattern] with [new element] 

 Additional Requirements: 
 The [original object/part/pattern] must already exist in the image. 
 It can be an entire object, a part of an object, a geometric shape, or a pattern. 
 The [new element] should clearly differ from the original and fit naturally into the image. 
 It can be another object, a different part, a new shape, text, or a new pattern. 
 Avoid replacing with intangible elements (e.g., gases, smoke, light, shadow). 
 Do not change colors — replacements must not involve altering the color of any existing element. 

 General Rules: 
 Keep the instruction short and clear. 
 No extra explanation or description. |
| Remove | Given an image, generate a short 'remove; type editing instruction in the format: Remove [object/part] 

 Additional Requirements: 
 The [object/part] must already exist in the image. 
 It can be the whole object or a specific part of an object (e.g., handle of a cup, branch of a tree). 
 The removal should be visually noticeable and affect the composition of the image. 
 Avoid removing intangible elements (e.g., light, shadow, gases, smoke). 

 General Rules: 
 Keep the instruction short and clear. 
 No extra explanation or description. |
| Add | Given an image, generate a short "add" type editing instruction in the format: Add [element] to [location] 

 Additional Requirements: 
 The [location] can be: 
 an existing object in the image, 
 a position within the image (e.g., top left, bottom center), 
 or a specific part/position of an object (e.g., handle of a cup, roof of a house). 
 The [element] should blend naturally into the image and not appear abrupt. 
 It can be an object, text, pattern, or other visual addition. 
 Avoid adding gases, smoke, or other intangible elements. 

 General Rules: 
 Keep the instruction short and clear. 
 No extra explanation or description. |

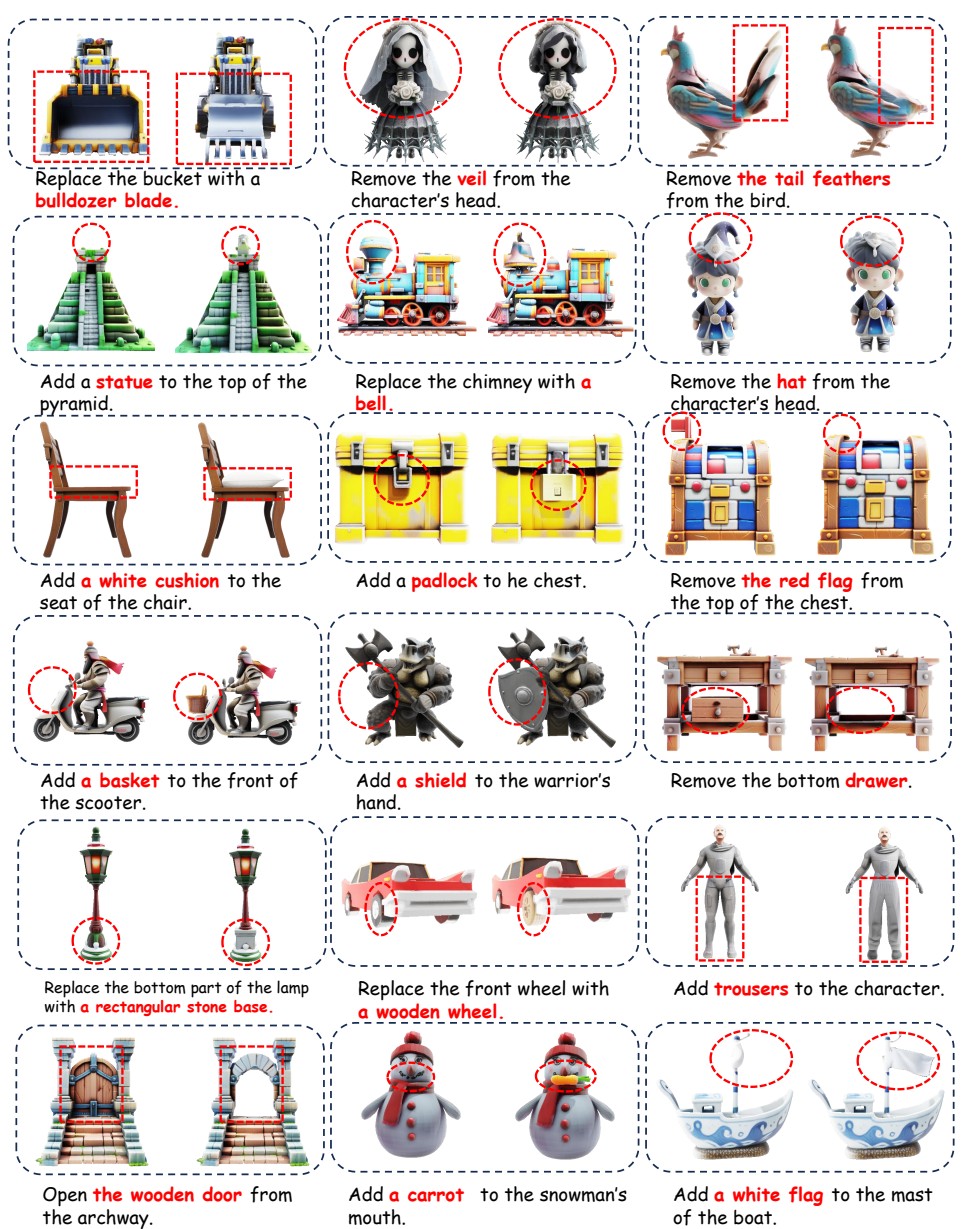

Figure 16: We present additional editing results involving addition, removal, and replacement. Edited regions are highlighted with red dashed circles. As shown, Nano3D achieves high editing consistency, preserving geometry and texture outside the edited areas.

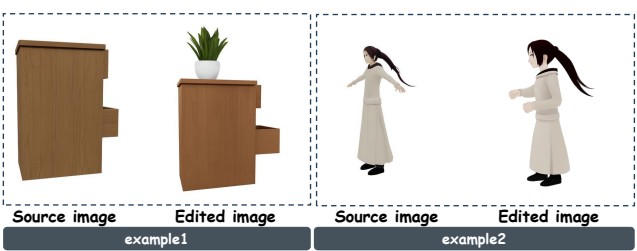

Figure 17: A bad case sampled from the 3D-Alpaca Ye et al. (2025) dataset shows that its image consistency before and after editing is poorly maintained.

