# OpenReview forum: "Nano3D: A Training-Free Approach for Efficient 3D Editing Without Masks"
_ICLR.cc/2026/Conference — ICLR 2026 Poster_

### Official Review · Reviewer_dUSo · 2025-10-31

**Soundness:** 3
**Presentation:** 3
**Contribution:** 3
**Rating:** 8
**Confidence:** 3

**Summary:**

Motivated by FlowEdit and TRELLIS, the authors propose Nano3D, a training-free framework for precise and coherent 3D object editing without requiring masks. The method first uses a closed-source image generation model to edit the front view of the input image (e.g., removing, adding, or replacing objects). It then performs 3D editing on both geometry and appearance, and leverages an XOR operation between sparse structures to effectively filter the regions that require editing. This pipeline ultimately achieves relatively high-fidelity 3D edits.

**Strengths:**

The paper is very clearly written, and the method is easy to understand (except for Section 4.2).

The visual results are impressive, significantly enhancing the credibility of Nano3D.

The authors curate the first large-scale 3D editing dataset, Nano3D-Edit-100k, which will be valuable to the research community.

The ablation studies thoroughly demonstrate the role of each component, and the contribution of each module is both intuitive and theoretically well-motivated.

**Weaknesses:**

The method relies on FlowEdit, which takes the front-view edited image as input. Does this imply that editing content on the back side of an object is not supported?

The authors use a fixed threshold τ = 100 to generate the edit mask. However, different editing tasks involve regions of vastly different scales—for example, a subtle edit like replacing human ears with elf ears should likely use a much smaller τ. This suggests that τ is a task-dependent hyperparameter requiring manual tuning, which limits practical usability.

The mask generation relies on geometry-induced changes in voxel occupancy. Consequently, the method cannot handle pure texture edits that preserve geometry (e.g., changing the color or pattern of an object without altering its shape).

Possibly due to my limited familiarity with FlowEdit, I found the descriptions in Sections 4.2 and 4.4—regarding the integration of FlowEdit and TRELLIS—to be overly brief, which created some difficulty in fully understanding the technical details.

**Questions:**

Please refer to the Weakness.

---

> ### Author Response · Authors · 2025-11-22
>
> **W1:  Support for Back-View and Occluded Region Editing.**
> We respectfully clarify that Nano3D supports editing on the back side and occluded regions. This capability is achieved through three complementary mechanisms:
>
> 1. Robustness to Partial Occlusion: As shown in the 4th column of Figure 5 (instruction: "Make the backpack the boy is carrying bigger"), Nano3D successfully enlarges the backpack even though it is not fully visible in the frontal view. This demonstrates that the model possesses a generative prior capable of inferring and editing partially occluded geometry.
> 2. Flexible Viewpoint Selection: The "front-view" input is a flexible user parameter rather than a fixed constraint. For editing tasks focused entirely on the back, the 3D object can simply be rotated (e.g., by 180 degrees) to expose the target region as the conditioning view.
> 3. Multi-View Guidance Mechanism: To address severe occlusions or simultaneous front-back editing, Nano3D supports multi-view conditioning. As illustrated in Figure 12, the system accepts both frontal and rear views (or additional views) simultaneously as input to FlowEdit. By ensuring the edited regions are covered by the guidance views, Nano3D effectively handles complex occlusion scenarios.
>
> **W2:  Robustness and Generalization of the Hyperparameter τ.**
> We thank the reviewer for this insightful observation regarding the limitation of a fixed threshold. To address this, we introduce an Adaptive Threshold Estimation algorithm in Appendix A.7, which automatically adjusts τ based on the granularity of the specific editing task. Specifically, the algorithm iteratively halves the threshold value if the connected components of potential edit regions are smaller than the current threshold, ensuring even fine-grained changes are captured. We validate this using the reviewer's specific example of replacing human ears with elf ears (see Figure 14); while the fixed threshold (τ=100) fails to detect the small ear regions, our adaptive approach automatically reduces τ to 25 to generate an accurate mask. This strategy effectively eliminates the need for manual hyperparameter tuning, ensuring robust performance across editing tasks of vastly different scales.
>
> **W3:  Clarification on Feasibility of Pure Texture/Appearance Editing.**
> We respectfully clarify that Nano3D supports pure texture editing without altering the underlying geometry. As detailed in the newly added Appendix A.2 and Figure 8, we achieve this through the following process: we first apply the desired texture edits to the rendered front-view image of the source mesh. Subsequently, we feed this edited image along with the original mesh into Trellis to produce textured Gaussians, and finally bake the resulting texture onto the input mesh. This procedure allows us to update the object's appearance (e.g., changing color or patterns) while strictly preserving its original geometric structure.
>
> **W4:  Elaboration on Technical Integration of FlowEdit and TRELLIS.**
> We sincerely apologize for the confusion. We recognize that the integration of FlowEdit within the TRELLIS framework is important, and the original description was indeed too brief for readers unfamiliar with these specific backbones. To address this, we have significantly revised the manuscript from two perspectives:
> 1. **Detailing the Integration Process.** We have completely rewritten Section 4.2 to include the necessary inputs, outputs, and the editing trajectory formulation on how TRELLIS utilize FlowEdit to perform 3D object editing, and this process can be summarized as two explicit stages:
>     (1) Front-view Image Editing. The input is the rendered front-view image of the source 3D object. Nano Banana is used to perform image editing, and the output is the front-view image of the target 3D object.
>     (2) Voxel Editing. Given the source 3D voxel grid as input, and both the source and target front-view images as conditions, TRELLIS applies FlowEdit aligorithms to output the edited 3D voxel grid.
> 2. **Providing Necessary Theoretical Background.** We have expanded Section 3 and Appendix A.12 to clarify TRELLIS’s 3D representation and its two-stage generation pipeline, as well as a more rigorous derivation of FlowEdit. Specifically, we detail how FlowEdit converts two independent noise-to-image generation trajectories into a direct training-free sourc image to target image trajectory.

---

> > ### Comment · Reviewer_dUSo · 2025-11-27
> >
> > Thank you for the authors' reply. My main concerns have been resolved, and I will maintain my rating.

---

> > > ### Author Response · Authors · 2025-11-27
> > >
> > > We are glad that your concerns have been addressed, and we would like to sincerely thank you for your thoughtful and constructive review.

---

### Official Review · Reviewer_zHpX · 2025-10-31

**Soundness:** 1
**Presentation:** 3
**Contribution:** 2
**Rating:** 4
**Confidence:** 4

**Summary:**

Nano3D is a training-free framework that enables precise and coherent 3D object editing without masks. It combines FlowEdit with TRELLIS for localized edits and introduces Voxel/Slat-Merge strategies to maintain structural consistency between edited and unedited regions.
Additionally, the authors present Nano3DEdit-100k, a large-scale dataset of over 100,000 high-quality 3D editing pairs, laying the foundation for future feed-forward 3D editing models.

**Strengths:**

[1] The paper clearly communicates its main claims and presents a valuable contribution in the form of zero-shot 3D editing.

[2] The quality of the presented results is also qualitatively impressive.

**Weaknesses:**

1. The proposed system shows a strong dependency on **FlowEdit** and **Trellis**, but the paper fails to clearly convey the fundamental nature of the models being utilized.
   - In this system, *FlowEdit* functions as an **image editor**. What matters is not merely that FlowEdit is used, but rather that **an image editing module** is employed — and its inputs and outputs should be clearly formulated.
   - The key insight readers should gain from this paper is the use of a **3D representation**, not simply the use of Trellis. Regardless of whether Trellis or another model is used, it is crucial to **define its input, output, and equations** clearly. In fact, in Section 3.2, there is only a brief mention of `z_i`, with no explanation of what `p_i` or `C` represent.

2. In Section 4.3, the explanation of **Voxel-Merge** is difficult to follow, as it assumes readers already know all the terms and omits formal definitions, which significantly reduces readability. Specifically, what does `s(i)` refer to?

3. **Voxel-Merge** is a proposed method, yet the explanation provided in Lines 203–210 is insufficient.

4. The claim that *"We obtain voxels that are fully consistent before and after editing"* is not a factual statement but rather an assertion. Isn’t the method still **dependent on thresholding**?

5. Overall, the paper gives the impression of **assembling multiple existing models** to perform a single task. I would encourage the authors to describe their ideas in a **more generalized and conceptual form**. As it stands, the paper reads more like a **technical report of an empirical study** than a research paper presenting a novel framework.

**Questions:**

My questions are in the weakness

**Details Of Ethics Concerns:**

No concerns

---

> ### Author Response · Authors · 2025-11-22
>
> **W1: Unclear Descriptions of FlowEdit and TRELLIS.**
> We sincerely appreciate the reviewer for pointing out the lack of the fundamental nature of the used models. To address these concerns, we have made the following revisions and clarifications:
> 1. **Formalizing the Editing Modules.** We have revised Section 4.1 to clearly outline the inputs, outputs, and critical equations of each model involved in FlowEdit for 3D object editing. In Appendix A.13, we have also added a description of FlowEdit’s underlying principles to provide the necessary background. The use of FlowEdit in the 3D object editing process can be summarized in two key modules:
>     (1) **Front-view Editing**: The input consists of the source 3D object's front-view image and an editing instruction. The Nano Banana model is used to perform the image editing, outputting the edited front-view image.
>     (2) **Voxel Editing**: In TRELLIS stage 1, the source 3D voxel grid is provided as input, with the source and modified front-view images serving as conditions before and after editing respectively, and the FlowEdit algorithm is then applied on TRELLIS to generate the edited 3D voxel grid.
> 2. **Clarifying the 3D Representation.** We have rephrased Section 3.2 to provide a clearer description of the two-stage generation process of TRELLIS, with explicit definitions of the inputs, outputs, and key representations. The TRELLIS process can be summarized as:
>     (1) **Stage 1 (Geometry Generation).** TRELLIS begins from noise, generating a voxel grid s, where s_i represents the binary state (occupied or empty) of the i-th voxel in the 3D grid.
>     (2) **Stage 2 (Appearance Generation).** TRELLIS further generates a structured latent (SLat) z on the generated voxel grid s in stage 1. Here, z_i is a vector of aggregated multi-view DINOv2 features corresponding to the i-th voxel, where the the dimensionality of the vector is C.
>
> **W2&W3: Unclear and Insufficient Explanations of Voxel-Merge.**
> We apologize for the confusion caused by the brevity of Voxel-Merge explanation. We make two efforts to address them:
> 1. **Regarding the notation s(i).** s(i) denotes the binary state (occupied or empty) of the i-th voxel in the 3D grid. Moreover, since most terms involved in Voxel-Merge come from from TRELLIS, we have provided their formal definitions in Section 3.2 to offer the necessary preliminary knowledge for reading Section 4.3.
> 2. **Regarding Voxel-Merge.** We have completely rewritten Section 4.3 to provide a more understandable and detailed explanation. The process of Voxel-Merge can be summarized as follows:
>     (1) **Input.**: The source voxel grid and the FlowEdit-edited voxel grid, which brings some undesirable small modifications to non-edited regions.
>     (2) **Difference Detection.**: We perform an element-wise XOR operation between the two grids to identify all voxels that have been modified.
>     (3) **Region Grouping.**: We apply connectivity analysis to group these modified voxels into distinct spatial regions.
>     (4) **Edited Region Selection.**: We set a threshold to filter out those modified non-edited regions and preserve the intentional edited regions.
>     (5) **Merging.** We adopt a mask to mark the selected regions and transfer them to the source voxel grid by another element-wise XOR operation with the mask.
>     (6) **Output.** The merged voxel grid with both the non-edited regions of the source voxel grid and edited regions of target voxel grid.
> This process ensures that the edited voxel retains sufficient geometry consistency with the source voxel grid.
>
> **W4: Overstatement on Fully Consistent Editing.**
> We thank the reviewer for the keen observation. We agree that the phrasing was slightly absolute. However, we clarify that empirically, our method achieves robust structural consistency in the vast majority of scenarios. As demonstrated in Figure 7 and Figure 14, with the implementation of both fixed and adaptive thresholding strategies, our pipeline effectively preserves non-edited regions in almost all tested cases. To ensure precision, we have revised the manuscript to use "highly consistent" instead of "fully consistent," reflecting our practical success without overclaiming.

---

> ### Author Response · Authors · 2025-11-22
>
> **W5: Clarification of Novelty and Conceptual Framework.**
> We understand the concern regarding the "assembly" of models. However, we respectfully argue that our contribution extends far beyond a simple combination of components. We have restructured the paper to highlight our conceptual contributions from two key perspectives:
>
> 1. Macro Perspective: Bridging the Gap in the 3D Editing Roadmap
> As shown in Section 1 introduction's paragraph 2 and 3, by drawing an analogy to 2D editing evolution, the development of a general 3D editing paradigm typically involves three stages: (1) training-free editing algorithms, (2) large-scale editing dataset construction, and (3) feedforward generalist editing models. Prior to our work, progress in the first two stages had been significantly constrained—there existed no consistent, training-free 3D editing algorithm and, consequently, no way to generate high-quality editing data at scale. Nano3D directly addresses this bottleneck. By resolving the core challenges of consistency and automation, Nano3D provides the first scalable data engine for 3D editing. This contribution effectively unlocks the possibility of pursuing large-scale, feedforward 3D editing models (Stage 3), which was previously infeasible due to the absence of data.
>
> 2. Micro Perspective: Technical Innovations and Empirical Insights
> From a technical standpoint, Nano3D is not a trivial assembly but is grounded in several key insights and dedicated modules designed to address fundamental 3D editing challenges. First, we provide a critical empirical finding by systematically demonstrating that FlowEdit is feasible when applied to SLAT representations, whereas it fails entirely on VecSet. This result offers an important guideline for future research on choosing representations suitable for training-free 3D editing. While we leverage FlowEdit (originally for 2D), applying it to 3D is non-trivial and prone to small undesirable modifications in non-edited regions. Building on this insight, we introduce two novel modules—Voxel-Merge and Slat-Merge—which adaptively determine editing regions without requiring any manual user input, such as mask specification. This capability, which was not achievable with prior methods, enables a fully automated and scalable editing pipeline. Consequently, we construct the first large-scale 3D editing dataset, providing a foundation that allows the community to scale 3D data generation in ways that were previously impossible.
>
> We have extensively rewritten Sections 3 and 4 to better articulate these system-level innovations and the logical cohesion between components, moving away from a "technical report" style to a unified research narrative.

---

> ### Author Response · Authors · 2025-11-27
>
> Thank you for your insightful comments and dedication to evaluating our manuscript. We have submitted a detailed rebuttal addressing all your points. We kindly request your feedback on our responses at your earliest convenience, and we would be pleased to discuss any points further with you. Your guidance remains invaluable as we refine our work, and we sincerely appreciate your time and expertise.

---

### Official Review · Reviewer_oayJ · 2025-10-31

**Soundness:** 2
**Presentation:** 2
**Contribution:** 3
**Rating:** 4
**Confidence:** 4

**Summary:**

This paper introduces a training-free 3D editing framework based on current powerful generative models like TRELLIS and Nano-Banana. In contrast to prior methods relying on 2D multi-view editing, they adapt FlowEdit into TRELLIS thus mitigating the problem of multi-view inconsistency to some extent. To enhance consistency, the authors propose Voxel/Slat-Merge strategy to preserve unedited regions via connectivity analysis. The framework enables a pipeline for constructing Nano3D-Edit-100k, the first large-scale 3D editing dataset. The paper provides evaluations on qualitative visuals, quantitative metrics (e.g., Chamfer Distance, DINO-I, FID), and user studies to demonstrate superior performance over baselines like Tailor3D, Vox-E, and TRELLIS. The work aims to advance 3D editing toward feedforward models, mirroring the evolution in 2D image editing.

**Strengths:**

1. **Stable and efficient data generation pipeline:** It effectively adapts FlowEdit from 2D to 3D editing within TRELLIS, demonstrating that pretrained generative priors can enable training-free, inversion-free edits.
2. **Significant Dataset Contribution:** Nano3D-Edit-100k fills a critical gap in 3D editing data, with automated construction using VLMs for diverse instructions and quality filtering. The comparison highlights the semantic alignment, making it a valuable resource for future supervised models.
3. **Comprehensive and extensive evaluation.**

**Weaknesses:**

1.  **Limited novelty**: it is appreciated to smartly integrate pretrained models into such framework. but the novelty is really limited, which introduces some concerns about the contribution to the research community.
2. **Limited edit operations**: now it only supports static add/removal/replacement, can’t support deformation and so on.
3.  **The claim of “without mask” is not convincing**. As in the voxel/slat-merging stage, it requires a difference map and a threshold to preserve original structure. This is actually still a mask, which could be easily got because the property of supported editing operations.

**Questions:**

1. Generally I like the whole pipeline. But why is the framework called “Nano3D”? What is the motivation behind this? I can’t really decode the meaning behind this. And I don’t think it’s appropriate to include Nano because of Nano-Banana.
2. How sensitive is Nano3D to the choice of front-view rendering? Could incorporating multiple views in FlowEdit guidance improve robustness for complex occlusions or asymmetric objects?
3. Nano3D-Edit-100k balances 10 categories, but what is the distribution of edit types (add/remove/replace) and instruction complexities? Could releasing per-category metrics help assess dataset quality?

**Details Of Ethics Concerns:**

None.

---

> ### Author Response · Authors · 2025-11-22
>
> **W1: Novelty and Contributions of Nano3D**
> We thank the reviewer for acknowledging our effective integration of pretrained models. However, we respectfully emphasize that the contribution of Nano3D extends substantially beyond a simple combination of existing components. Our work introduces both conceptual and technical advances that we believe are important to the 3D editing community. We summarize these contributions from two complementary perspectives:
>
> 1. Macro Perspective: Establishing a Roadmap for General 3D Editing
> Analogous to the evolution of 2D image editing, the development of a general 3D editing paradigm typically involves three stages: (1) training-free editing algorithms, (2) large-scale editing dataset construction, and (3) feedforward generalist editing models. Prior to our work, progress in the first two stages had been significantly constrained—there existed no consistent, training-free 3D editing algorithm and, consequently, no way to generate high-quality editing data at scale. Nano3D directly addresses this bottleneck. By resolving the core challenges of consistency and automation, Nano3D provides the first scalable data engine for 3D editing. This contribution effectively unlocks the possibility of pursuing large-scale, feedforward 3D editing models (Stage 3), which was previously infeasible due to the absence of data.
>
> 2. Micro Perspective: Technical Innovations and Empirical Insights
> From a technical standpoint, Nano3D is not a trivial assembly but is grounded in several key insights and dedicated modules designed to address fundamental 3D editing challenges. First, we provide a critical empirical finding by systematically demonstrating that FlowEdit is feasible when applied to SLAT representations, whereas it fails entirely on VecSet. This result offers an important guideline for future research on choosing representations suitable for training-free 3D editing. While we leverage FlowEdit (originally for 2D), applying it to 3D is non-trivial and prone to small undesirable modifications in non-edited regions. Building on this insight, we introduce two novel modules—Voxel-Merge and Slat-Merge—which adaptively determine editing regions without requiring any manual user input, such as mask specification. This capability, which was not achievable with prior methods, enables a fully automated and scalable editing pipeline. Consequently, we construct the first large-scale 3D editing dataset, providing a foundation that allows the community to scale 3D data generation in ways that were previously impossible.
>
> **W2: Limited edit operations**
> We thank the reviewer for the feedback. We respectfully clarify that Nano3D is capable of performing deformation-based editing in addition to static add/removal/replacement operations. To address this concern, we have included additional experimental results in Appendix A.3. As illustrated in the newly added Figure 9, our method successfully handles deformation tasks and achieves highly consistent results across different views.
>
> **W3: The claim of 'without mask' is not convincing.**
> We respectfully clarify that our claim of "without mask" specifically refers to the elimination of manual user intervention (e.g., user-drawn masks or bounding boxes), rather than the absence of internal masking operations. This distinction is fundamental for two reasons:
> Automation vs. Manual Labor: While the reviewer is correct that we utilize difference maps internally to preserve structure, this process is fully automatic and adaptive. Unlike existing methods that heavily rely on users to manually specify editing regions—a tedious and non-scalable process—Nano3D requires no human input to define where to edit.
> Enabling Scalable Data Construction: The removal of the manual masking requirement is precisely what makes our pipeline scalable. Because Nano3D operates without user-provided bounding boxes or masks, it enables the automated construction of 3D editing datasets at the scale of millions. This capability for large-scale data generation is the most significant advantage of our method and is unattainable by approaches that depend on manual guidance.

---

> ### Author Response · Authors · 2025-11-22
>
> **Q1:Motivation behind the Framework Name and Clarification**
> We appreciate the reviewer’s feedback regarding the framework's name. We chose "Nano3D" not merely as a reference, but to reflect three key aspects of our method's design and vision:
> 1. **Direct Technical Integration.** Our pipeline explicitly leverages Nano-Banana for the initial 2D image editing stage. As a state-of-the-art image editing model, Nano-Banana provides the high-quality visual guidance that powers our 3D consistency. Including "Nano" in the name transparently acknowledges this critical component of our system.
>
> 2. **Bridging the Gap in 3D Editing.** Nano-Banana represents a milestone in 2D editing, characterized by its powerful instruction following and ease of use; Nano3D aims to bring this exact level of capability to the 3D domain. Unlike previous 3D editing methods that heavily relied on[1] bounding boxes (BBox) as constraints—which severely limited the scalability of data construction—our framework introduces a novel, training-free approach that eliminates these constraints. This effectively bridges the gap between 2D editing capabilities and 3D structural consistency, advancing 3D editing to a new stage of accessibility and scalability.
>
> 3. **Vision for the Future.** Finally, the name reflects our aspiration. Just as Nano-Banana revolutionized 2D image manipulation, we envision Nano3D as a foundational step toward achieving a "Nano-Banana level" of proficiency in 3D—where users can edit complex 3D assets with the same freedom and precision currently available for 2D images.
>
> **Q2:Sensitivity and Multi-view Guidance**
> We thank the reviewer for these insightful questions. We address them in two parts:
>
> 1. Sensitivity to Front-View Selection: We investigate the impact of the initial viewpoint in Appendix A.5 and Figure 13. Our ablation study applies Nano3D to four distinct viewpoints of a fixed mesh. The results demonstrate that our method is robust to viewpoint variations: it yields highly coherent outcomes across side-oriented views and produces plausible edits even when initialized from a fully rear-facing perspective (despite minor consistency reductions).
> 2. Multi-View Guidance for Robustness: We validate that incorporating multiple views significantly improves robustness for complex occlusions. In Appendix A.6 and Figure 12, we introduce a multi-view guidance mechanism where edited images from both frontal and rear views are jointly injected into the FlowEdit pipeline. This strategy successfully enables the simultaneous editing of spatially distinct features (e.g., adding a fist on the front and a backpack on the back) in a single-pass process, effectively handling asymmetric objects.
>
> **Q3:**
> We thank the reviewer for this insightful suggestion regarding the dataset statistics and fine-grained evaluation. In the revised manuscript, we have added a detailed analysis in Appendix A.8 to address these points.
> 1. Distribution of Edit Types:
> As shown in Table 4 of the revised paper, we provide the distribution of the three editing operations. The dataset maintains a balanced composition across Add, Remove, and Replace instructions to ensure model robustness across different editing scenarios.
>
> 2. Distribution of Instruction Complexity:
> In Table 5, we present the distribution of instruction difficulties. To ensure an objective assessment, we employed a Vision-Language Model (VLM) to score the complexity of each instruction on a scale of 0 to 5. The prompt used for this evaluation is as follows:
> You are an evaluator for 3D object editing tasks. Assess the difficulty of the following instruction on a scale from 0 to 5:
> Instruction: {instruction}
> Output strictly only the single digit.
> The results indicate that the majority of instructions fall within the moderate difficulty range (Score 3), while retaining a reasonable proportion of simple and complex instructions to simulate real-world usage.
>
> 3. Per-Category Metrics:
> We agree that global metrics might mask category-specific performance nuances. Therefore, we have released the per-category semantic alignment evaluation in Table 6.
> The results reveal that our method achieves highly stable and robust performance on object-centric categories (e.g., Food, Personal Item). However, the performance is relatively lower on categories with high visual ambiguity or structural complexity, such as Weapon, Building, and Plant. We have included a discussion in the revised paper acknowledging these challenges, which points to meaningful directions for future optimization in fine-grained semantic alignment.

---

> ### Author Response · Authors · 2025-11-27
>
> Thank you for your insightful comments and dedication to evaluating our manuscript. We have submitted a detailed rebuttal addressing all your points. We kindly request your feedback on our responses at your earliest convenience, and we would be pleased to discuss any points further with you. Your guidance remains invaluable as we refine our work, and we sincerely appreciate your time and expertise.

---

### Official Review · Reviewer_wt7j · 2025-11-03

**Soundness:** 3
**Presentation:** 3
**Contribution:** 3
**Rating:** 6
**Confidence:** 3

**Summary:**

This paper proposes Nano3D, a training-free 3D object editing pipeline that plugs FlowEdit into TRELLIS’s first-stage voxel generator and then preserves untouched content via region-aware merging at both voxel and SLAT levels (Voxel-Merge / SLat-Merge). Nano3D reports the best quantitative performance among baselines. A user study (n = 50) also found that participants prefer Nano3D across prompt alignment, visual quality, and shape preservation.  Lastly, the paper further introduces Nano3D-Edit-100k, a 100k-pair dataset built with an automated pipeline.

**Strengths:**

* **Simple and effective approach**: This paper mainly focuses on keeping non-edited regions, and the proposed approach is simple but effective. Compared to directly extending FlowEdit to TRELLIS, analyzing issues (such as changing connected regions) makes sense, and the Voxel & SLat merge helps to resolve this.
* **Impressive empirical results**: Nano3D leads on CD/DINO-I/FID (Table 1) and is preferred in the user study (Table 2); ablations show Voxel-Merge and SLat-Merge contribute additively. From the qualitative results, the non-edited parts are clearly maintained without harming the generation quality.
* **3D Editing Dataset**: They specify a practical 100k-pair pipeline, achiving better alignment than 3D-Alpaca. Such a dataset and tool may be helpful for future work.
* The overall writing and illustrations are clear to follow and understand.

**Weaknesses:**

* The method uses "input 3D object’s front view" as a source condition. Thus, for the edits on the unseen part, it may lead to problems. In addition, while merging helps preserve the rest, the 3D editing quality may hinge on the quality of that 2D edit. It is good to see that the paper itself notes many failures originate in the 2D editing stage, but there’s no controlled study of viewpoint sensitivity (e.g., edits specified from non-frontal views or multi-view edits). A better analysis of the method's robustness will help to understand the boundary of the method.
* The method highly relies on the Trellis representation, which makes it hard to extend. Specifically, the input must be the object style of Trellis and is generalizable to complex scenes or objects.
* The compared baselines are too old and narrow. For example, Vox-E is a work from ICCV2023. For an informative comparison, more up-to-date works should be considered, such as PrEditor3D (CVPR 2025) or recent works.

**Questions:**

* In Figure 5, the scale of the rendered object seems to change compared with Trellis (smaller). Is it because of the method or the plotting issue? In addition, why does Trellis show some degeneration on the objects compared with Nano3D?

---

> ### Author Response · Authors · 2025-11-22
>
> **W1: Analysis of Viewpoint Sensitivity and Robustness.**
> We thank the reviewer for the insightful comments regarding the method's robustness to viewpoint selection and unseen regions. We have conducted additional analyses and experiments to address these concerns:
> 1. **Robustness to Unseen Parts and Occlusions.** Editing unseen or occluded parts is feasible within our framework. As demonstrated in Figure 5 (4th column), for the prompt "Make the backpack the boy is carrying bigger," the method successfully enlarges the backpack even though it is largely occluded in the frontal view. This indicates the model's capability to handle partial occlusions based on semantic context. Furthermore, for severe occlusions, our method offers flexible solutions: the input object can be rotated to use a better viewpoint (e.g., back view), or multiple views (e.g., front and back) can be used simultaneously as conditions, as shown in Figure 12. This ensures that occluded regions can be explicitly covered when necessary.
> 2. **Controlled Study on Viewpoint Sensitivity.** To explicitly analyze viewpoint sensitivity, we performed a controlled study (added in Figure 13) where edits were generated from 4 independent viewpoints (including non-frontal and back views). The results show that Nano3D successfully performs the edits across all tested angles. Although the consistency of the back-view edit is slightly lower than that of the frontal view (likely due to the 2D prior's bias), the overall success demonstrates that our method is robust to viewpoint variations and is not strictly limited to frontal inputs.
>
> **W2: Discussion on Generalizability and Extensibility.**
> We acknowledge that our method is built upon the SLAT representation from TRELLIS. However, we respectfully clarify that this dependence does not hinder the method's extensibility or limit it solely to object-level editing. As detailed in the new Scene Editing section, our method demonstrates strong generalizability:
> 1. **Direct Generalization to Scenes.** Nano3D is not confined to single objects. For scenes with moderate complexity, the SLAT representation is sufficient to encode the geometry and texture. As shown in Figure.10, Nano3D can be directly applied to edit entire indoor and outdoor scenes (e.g., changing the style or layout of a room), demonstrating its native capability beyond simple object editing.
>
> 3. **Extensibility to Complex Scenes via Block-wise Strategy.** To address the limitation where highly complex scenes exceed the capacity of a single SLAT encoding, we introduce a scalable Block-wise Editing Strategy. Specifically, instead of processing the entire scene at once, we partition it into manageable blocks and apply Nano3D exclusively to the block containing the target region. This "divide-and-conquer" approach effectively bypasses the representation bottleneck. As demonstrated in Figure 11, this strategy enables us to perform hierarchical edits on a city-scale scene—first replacing a building with a garden, and subsequently transforming a fountain within that garden into a sculpture. These results confirm that our method is not limited to simple objects and can be effectively extended to complex environments.
>
> **W3: Additional Comparisons with Up-to-Date Baselines.**
> To ensure a comprehensive evaluation, we have updated the paper with the following comparisons:
> 1. Comparison with Recent Baselines. We have added a comparative evaluation with [Instant3DiT] in Table 1. The results further validate the effectiveness of Nano3D in maintaining high-quality geometry and texture during editing.
>
> 2. Since the official source code for PrEditor3D is currently unavailable, we were unable to reproduce it directly on our dataset. Instead, to ensure a fair assessment, we performed a qualitative comparison using the official examples provided on their project page (see Figure 15). Visual inspection reveals that PrEditor3D struggles to preserve the integrity of non-edited regions; specifically, in the "Car" example, the shape and texture of the side mirror are unintentionally altered during the editing process. In stark contrast, Nano3D successfully executes the target edit while maintaining near-perfect consistency in the surrounding areas, including the side mirror. This comparison highlights our method's superior capability in disentangling the edited region from the original structure, ensuring high-fidelity content preservation.

---

> ### Author Response · Authors · 2025-11-22
>
> **Q1: Clarification on Visualizations in Figure 5.**
> Regarding the observations in Figure 5, we clarify that the perceived change in scale is primarily a plotting issue caused by the auto-normalization of camera distance based on the updated bounding box during rendering, rather than an actual geometric distortion by our method. As for the structural degradation observed in the Trellis baseline, this arises because Trellis lacks the Voxel Merge and SLAT Merge operations that are integral to Nano3D. Without these structural constraints, the generation process in Trellis becomes over-dependent on the target condition, causing the optimization to drift away from the original geometry. This inability to anchor the edit to the source structure leads to a loss of consistency, resulting in the visual degeneration seen in the comparison.

---

> ### Author Response · Authors · 2025-11-27
>
> Thank you for your insightful comments and dedication to evaluating our manuscript. We have submitted a detailed rebuttal addressing all your points. We kindly request your feedback on our responses at your earliest convenience, and we would be pleased to discuss any points further with you. Your guidance remains invaluable as we refine our work, and we sincerely appreciate your time and expertise.

---

### Author Response · Authors · 2025-11-26

Dear Reviewers,

With the discussion period ending soon, we kindly invite you to join the discussion so that we have enough time to address any remaining concerns in a careful and thorough way. If there is anything you would like us to clarify, please let us know and we will respond promptly.

Thank you very much for your time and consideration.

---

### Comment · Area_Chair_aNwL · 2025-11-26
**Gentle Reminder: Response to Authors’ Rebuttal**

Dear Reviewers,

This is a gentle reminder to please review and respond to the authors’ comments at your earliest convenience. Your engagement during the rebuttal phase is essential to ensuring a fair and thorough evaluation process.

The deadline for responding to author comments is **December 2**.

We appreciate your effort and dedication to maintaining the quality of the review process.

Thank you,

AC

---

### Author Response · Authors · 2025-12-03
**Summary for Area Chair**

Dear Area Chair,

We sincerely appreciate that, in such an unexpected and challenging situation, you have still generously devoted your valuable time to thoroughly evaluating our submission and rebuttal.

To assist your evaluation, we provide a brief summary below.

---
**Paper Overview**
* This paper presents **Nano3D**, a training-free, mask-free, and scalable 3D asset editing framework. Compared to previous approaches, Nano3D integrates FlowEdit into TRELLIS and introduces **Voxel/SLat-Merge** modules, achieving semantically accurate modifications while sufficiently preserving the geometry and appearance consistency of non-edited regions. Building on Nano3D, a **data construction pipeline** is further proposed, producing the first large-scale and high-quality 3D editing dataset **Nano3D-Edit-100K**. Together, Nano3D fills the blank of the first two components in 3D editing development path, like that of 2D image editing (**training-free algorithm** → **large-scale dataset** → specialized model), laying the foundation for training futural feedforward 3D editing models.

**Initial Ratings**
* 8 4 4 6 (Avg: 5.5).

**Recognized Strengths by All Reviewers**
* Simple, effective, and principled framework.
* Clear presentation and impressive empirical results.
* Significant 3D editing dataset contribution.

**Existing Discussions with Reviewers**
* **Reviewer dUSo:** *"Thank you for the authors' reply. **My main concerns have been resolved,** and I will maintain my rating (8→8)."*

We regret that **the other three reviewers (zHpX, oayJ, wt7j)** did not participate in the discussion before the incident happened. However, we have provided detailed responses and additional experiments addressing all the concerns and questions raised in their initial reviews.

**Main Concerns and Questions Resolved:**
| **Issue** | **Solution** |
| --------- | ------------ |
|**Baselines&Dataset Evaluation** | Added comparisons with **Instant3DiT** (Tab. 1) and **PrEditor3D** (Fig. 15) to demonstrate Nano3D achieves better performance. Clarified visual artifacts in Fig. 5. (Author Comment on Reviewer wt7j Q1). Released detailed **dataset statistics** (Tab. 4-6) to prove its quality and diversity. |
| **Robustness&Generalization** | Conducted experiments to demonstrate Nano3D achieves: (1) **Viewpoint Robustness** (Fig. 12-13); (2) **Scene Editing** (Fig. 10-11); (3) **Deformation/Texture Editing** (Fig. 8-9). |
| **Technical Refinements&Clarity** | Introduced **Adaptive Threshold Estimation algorithm** (Appendix A.7) for fine-grained editing (Fig. 14). Rewrote Sections 3&4 and added Appendix A.13 to provide a systematic explanation of the method. Clarified the assertion "fully consistent" (Author Comment on Reviewer zHpX W4). |
| **Novelty&Contribution** | Emphasized the contributions of Nano3D from three perspectives: **strategic roadmap, empirical insights, and technical innovations.** (Author Comment on Revierwer oayJ W1). Clarified the definitions and origins of term "mask-free" and "Nano3D" (Author Comment on Reviewer oayJ W3&Q1).

We also appreciate the constructive feedback from all reviewers, and we believe that our revisions, new experiments, and detailed responses have sufficiently addressed all the issues they presented. We kindly hope you can consider our rebuttal when making your decision, and once again, thank you so much for your efforts to the review process during this difficult period.

Yours sincerely,
The Authors

---

### Meta-Review · Area_Chair_h6w6 · 2026-01-08

**Summary:**

The paper proposes Nano3D, a training-free framework for 3D object editing that integrates FlowEdit into the TRELLIS generation pipeline. The method introduces "Voxel-Merge" and "SLat-Merge" strategies to preserve the consistency of unedited regions without requiring manual masks. Furthermore, the authors utilize this framework to construct and release "Nano3D-Edit-100k," a large-scale dataset comprising over 100,000 3D editing pairs.
Initial concerns focused on the method's novelty, the clarity of technical descriptions, and robustness regarding viewpoint sensitivity and baselines:
1. Reviewers oay and zHpX pointed out that the method lacks sufficient novelty, as it appears to be a simple assembly of existing models (FlowEdit and TRELLIS) and reads more like a technical report than a research paper.
2. Reviewers wt7j and dUSo raised concerns about the robustness of the method regarding viewpoint selection, specifically asking if it can handle back-view editing or occluded regions since it relies on front-view rendering.
3 . Reviewer oay argued that the claim of being "mask-free" is not convincing, because the Voxel-Merge strategy essentially generates an internal mask using a difference map and thresholds.
4. Reviewer wt7j noted that the baselines used for comparison (like Vox-E) are outdated and suggested comparing with more recent works like PrEditor3D.
5 . Reviewer zHpX criticized the clarity of the presentation, stating that the formal definitions of FlowEdit inputs/outputs and the explanation of the Voxel-Merge mechanism were insufficient.

The authors provided a comprehensive rebuttal that introduced an adaptive thresholding algorithm, added comparisons with state-of-the-art baselines like Instant3DiT and PrEditor3D, and formalized the methodology sections. Although three reviewers did not participate in the final discussion, the provided evidence sufficiently resolves the objective technical flaws raised during the review process.

**Reviewer Concerns:**

1. Methodology Clarity & Definitions. Reviewers requested formal definitions for the FlowEdit-TRELLIS integration and Merge modules, while challenging the "mask-free" terminology by arguing that internal heuristic thresholds constitute implicit masking. The authors revised Sections 3 and 4 to provide formal definitions and detail the integration process. They clarified that "mask-free" specifically refers to the elimination of manual user intervention rather than internal adaptive operations. While the presentation has improved significantly, the reliance on heuristic thresholds validates the reviewer's point that the method retains implicit masking logic.
2. Sufficiency of Baselines. Reviewers noted the absence of key baselines such as Instant3DiT and PrEditor3D, requesting comparisons to establish the method's superiority. The authors included comparisons with Instant3DiT and provided qualitative evaluations against PrEditor3D, noting that quantitative comparison was impossible due to code unavailability. The additional comparisons adequately contextualize the method's performance given the available resources.
3. Viewpoint Robustness & Generalization. Concerns were raised regarding the method's sensitivity to front-view selection and its capability to handle occlusions during the editing process. The authors introduced an ablation study on front-view selection and implemented a multi-view guidance mechanism to address occlusion issues. The additional experiments demonstrate satisfactory robustness and generalization capabilities.
4. Technical Novelty. Reviewers questioned the fundamental algorithmic novelty, characterizing the approach as an engineering combination of pre-trained models rather than a significant theoretical advancement. The authors emphasized the technical challenges in adapting FlowEdit to SLAT representations and highlighted the contribution of the large-scale Nano3D-Edit-100k dataset. Although the core algorithmic novelty is limited to engineering integration, the contribution of the first large-scale editing dataset and the method's superior empirical efficiency provide significant practical value to the community.

**Reviewer Scores:**

- Reviewer wt7j (Original: 6): The rebuttal addressed the concerns regarding baselines and viewpoint robustness. In the absence of further engagement, the score will likely solidify at 6 or arguably rise to 8.
- Reviewer oay (Original: 4): While the authors clarified the "mask-free" definition (automated vs. manual), the concern regarding "limited novelty" remains subjective. The score likely remains at 4 or marginally rises to 6 based on the dataset's value.
- Reviewer zHpX (Original: 4): The low "Soundness" rating stemmed from unclear definitions and writing style, which the authors extensively rewrote. With the comprehension barrier removed, the score is expected to rise to 6.
- Reviewer dUSo (Original: 8): The reviewer explicitly confirmed that the adaptive threshold solution resolved their concerns and maintained their score of 8

---

### Decision · Program_Chairs · 2026-01-26

Accept (Poster)